



# Antarctic Bottom Water and North Atlantic Deep Water in CMIP6 models

Céline Heuzé[1]

[1]Department of Earth Sciences, University of Gothenburg, Gothenburg, Sweden

**Correspondence:** Céline Heuzé (celine.heuze@gu.se)

**Abstract.** Deep water formation is the driver of the global ocean circulation, yet it was poorly represented in the previous generation of climate models. We here quantify biases in Antarctic Bottom Water (AABW) and North Atlantic Deep Water (NADW) formation, properties, transport and global extent in 35 climate models that participated in the latest Climate Model Intercomparison Project (CMIP6). Several CMIP6 models are correctly forming AABW via shelf processes, but in both hemi-
spheres, the large majority of climate models form deep water via open ocean deep convection, too deep, too often, over too large an area. Models that convect the least form the most accurate AABW, but the least accurate NADW. The four CESM2 models with their "pipe" / overflow parameterisation are among the most accurate models. In the Atlantic, the colder AABW, the stronger the abyssal overturning at 30°S, and the further north the AABW layer extends. The saltier NADW, the stronger the Atlantic Meridional Overturning Circulation (AMOC), and the further south the NADW layer extends. In the Indian and
Pacific oceans in contrast, the fresher models are the ones who extend the furthest regardless of the strength of their abyssal overturning, most likely because they also are the models with the weakest fronts in the Antarctic Circumpolar Currents. There are clear improvements since CMIP5: several CMIP6 models correctly represent or parameterise Antarctic shelf processes, fewer models exhibit Southern Ocean deep convection, more models convect at the right location in the Labrador Sea, bottom density biases are reduced, and abyssal overturning is more realistic. But more improvements are required, e.g. by generalising
the use of overflow parameterisations or by coupling to interactive ice sheet models, before deep water formation, and hence heat and carbon storage, are represented accurately.

## 1 Introduction

At both poles, deep water formation sets the global ocean ventilation in motion. Ocean-ice-atmosphere interactions by the
Antarctic ice shelves (Orsi, 2010; Drucker et al., 2011; Ohshima et al., 2013) or, more rarely, in open ocean polynyas (Killworth, 1983; Campbell et al., 2019), create the coldest, densest deep water: the Antarctic Bottom Water (AABW). AABW does not stay around Antarctica, but rather travels north on the sea floor as a 2000 m thick layer, filling all three basins (Johnson, 2008). In the Atlantic, the progression of AABW is stopped by the other deep water (Johnson, 2008), the North Atlantic Deep Water



(NADW). NADW forms in the Labrador Sea and Nordic Seas because of strong winds and haline convection respectively
(Killworth, 1983). It is the saltiest of the two water masses, but is also warmer and lighter than AABW and hence quits the sea
floor to continue circulating above AABW when the two meet (Johnson, 2008). NADW production has long been linked to
the strength of the Atlantic Meridional Overturning Circulation (AMOC, e.g. Broecker, 1995), although observations from the
recently deployed OSNAP line (Lozier et al., 2019) reveal that it is more complex than "more deep water formation = stronger
AMOC". More crucially maybe, AABW and NADW formation provide a direct path from the atmosphere to the bottom of the
ocean, and as such, a conduit for heat and carbon storage (Chen et al., 2019; Zanna et al., 2019). An accurate representation of
deep water formation in climate models is thus a necessary precondition for trustworthy future climate predictions.

The Climate Model Intercomparison Project phase 6 (CMIP6, Eyring et al., 2016) is the latest release of CMIP, the organised
effort to make global climate models comparable, notably by running them with the same forcings. In the previous installment,
CMIP5 (Taylor et al., 2012), Southern Ocean mixed layers were poorly represented (Sallée et al., 2013) and models were
forming the majority of their AABW wrongly via open ocean convection, mostly in the occurring-too-often Weddell Polynya
(Heuzé et al., 2013). They would only stop doing so once the ocean surface had freshened enough, by 2200 (De Lavergne et al.,
2014) and, consequently, underestimated 21st century bottom property changes (Heuzé et al., 2015). NADW formation was
more accurately represented, although due to too large sea ice extents in the North Atlantic, it occurred more in the Irminger
Sea than in the Labrador Sea (Menary et al., 2015; Heuzé, 2017). So far, results on CMIP6 models have shown that sea ice
representation has improved in both hemispheres, but the intermodel spread remains large (Roach et al., 2020; Shu et al.,
2020). The Weddell Polynya is still opening too often, but only in half of the models (Mohrmann et al., subm.)... those that
now have the most accurate Antarctic Circumpolar Current (Meijers et al., 2012; Beadling et al., 2020). CMIP6 models also
have a higher climate sensitivity than CMIP5 models (Zelinka et al., 2020), more in line with the observed sensitivity (Armour,
2017; Cox and Williamson, 2018). Their AMOC however is too sensitive to the new aerosol forcings (Menary et al., 2020).
CMIP6 resolution is still coarse, with most models having a horizontal resolution of 1°, but recent results showed that NADW
formation is in fact less accurate with higher resolution (Koenigk et al., 2020). In summary, by improving these crucial other
processes but keeping a low resolution, CMIP6 models should have more realistic AABW and NADW than CMIP5 models.
We here investigate whether this is the case.

In this paper, we determine the characteristics of Antarctic Bottom Water (section 3.1) and North Atlantic Deep Water
(section 3.2) in CMIP6 models, focussing first on their respective formation processes, properties, and biases. We then study
the global transport of these two water masses (section 3.3), in particular, how their properties determine their global extent.
Finally (section 4), we conclude this paper by a discussion on what -if anything- has improved since CMIP5.

## 2 Methods

### 2.1 CMIP6 models and observation-based reference data

We use the 35 CMIP6 models listed in Table 1. The only criterion for choosing them was the availability of at least their
seawater salinity and temperature monthly output "so" and "thetao" for the entire historical run (January 1850 to December





**Table 1.** The 35 CMIP6 models used in this study; their ocean component; horizontal resolution in ° latitude x ° longitude; vertical grid type ($\rho$ means isopycnic, $\sigma$ terrain-following, several symbols a hybrid grid) and number of vertical levels; and official reference. N/A indicates that no paper has been published yet for the CMIP6 configuration.

|   | Model name | Ocean component | Horizontal | Vertical | Reference |
|---|---|---|---|---|---|
| 1 | ACCESS-CM2 | MOM5 | 1 x 1 | z* 50 | N/A |
| 2 | ACCESS-ESM1-5 | MOM5 | 1 x 1 | z* 50 | Ziehn et al. (2017) |
| 3 | BCC-CSM2-MR | MOM4-L40 | 1 x 1 | z 40 | Wu et al. (2019) |
| 4 | BCC-ESM1 | MOM4-L40 | 1 x 1 | z 40 | Wu et al. (2019) |
| 5 | CAMS-CSM1-0 | MOM4 | 1 x 1 | z 50 | Rong et al. (2019) |
| 6 | CESM2 | POP2 | 1 x 1 | z 60 | Danabasoglu et al. (2020) |
| 7 | CESM2-FV2 | POP2 | 1 x 1 | z 60 | Danabasoglu et al. (2020) |
| 8 | CESM2-WACCM | POP2 | 1 x 1 | z 60 | Danabasoglu et al. (2020) |
| 9 | CESM2-WACCM-FV2 | POP2 | 1 x 1 | z 60 | Danabasoglu et al. (2020) |
| 10 | CNRM-CM6-1 | NEMO3.6 | 1 x 1 | z* 75 | Voldoire et al. (2019) |
| 11 | CNRM-ESM2-1 | NEMO3.6 | 1 x 1 | z* 75 | Séférian et al. (2019) |
| 12 | CanESM5 | NEMO3.4.1 | 1 x 1 | z 45 | Swart et al. (2019) |
| 13 | EC-Earth3 | NEMO3.6 | 1 x 1 | z* 75 | N/A |
| 14 | EC-Earth3-Veg | NEMO3.6 | 1 x 1 | z* 75 | N/A |
| 15 | GFDL-CM4 | MOM6 | 0.25 x 0.25 | $\rho$ - z* 75 | Held et al. (2019) |
| 16 | GFDL-ESM4 | MOM6 | 0.5 x 0.5 | $\rho$ - z* 75 | N/A |
| 17 | GISS-E2-1-G | GISS Ocean | 1.25 x 1 | z 40 | N/A |
| 18 | GISS-E2-1-G-CC | GISS Ocean | 1.25 x 1 | z 40 | N/A |
| 19 | GISS-E2-1-H | HYCOM | 1 x 1 | z - $\rho$ - $\sigma$ 32 | N/A |
| 20 | HadGEM3-GC31-LL | NEMO-HadGEM3-GO6.0 | 1 x 1 | z* 75 | Kuhlbrodt et al. (2018) |
| 21 | INM-CM5-0 | INM-OM5 | 0.5 x 0.25 | $\sigma$ 40 | Volodin and Gritsun (2018) |
| 22 | IPSL-CM6A-LR | NEMO3.6 | 1 x 1 | z* 75 | Lurton et al. (2020) |
| 23 | MCM-UA-1-0 | MOM1 | 2 x 2 | z 18 | N/A |
| 24 | MIROC-ES2L | COCO4.9 | 1 x 1 | z - $\sigma$ 62 | Hajima et al. (2020) |
| 25 | MIROC6 | COCO4.9 | 1 x 1 | z - $\sigma$ 62 | Tatebe et al. (2019) |
| 26 | MPI-ESM-1-2-HAM | MPIOM1.6.3 | 1.5 x 1.5 | z 40 | Mauritsen et al. (2019) |
| 27 | MPI-ESM1-2-HR | MPIOM1.6.3 | 0.4 x 0.4 | z 40 | Müller et al. (2018) |
| 28 | MPI-ESM1-2-LR | MPIOM1.6.3 | 1.5 x 1.5 | z 40 | Mauritsen et al. (2019) |
| 29 | MRI-ESM2-0 | MRI.COM4.4 | 1 x 0.5 | z* 60 | Yukimoto et al. (2019) |
| 30 | NESM3 | NEMO3.4 | 1 x 1 | z 46 | Cao et al. (2018) |
| 31 | NorCPM1 | MICOM | 1 x 1 | z - $\rho$ 53 | Counillon et al. (2016) |
| 32 | NorESM2-LM | MICOM | 1 x 1 | z - $\rho$ 53 | Tjiputra et al. (2020) |
| 33 | NorESM2-MM | MICOM | 1 x 1 | z - $\rho$ 53 | Tjiputra et al. (2020) |
| 34 | SAM0-UNICON | POP2 | 1 x 1 | z 60 | Park et al. (2019) |
| 35 | UKESM1-0-LL | NEMO-HadGEM3-GO6.0 | 1 x 1 | z* 75 | Sellar et al. (2020) |

2014) at the latest date of download (20 May 2020). When available, we also directly used their monthly mixed layer depth "mlostst"; if not, we computed it from the monthly salinity and temperature as detailed in section 2.2. We also made use of each model's bathymetry "deptho" and grid cell area "areacello" files to accelerate our computations. Finally, for the transports

60  calculations, we used the monthly meridional velocity "vo". All output were obtained on the model's native grid, except for NorESM2-LM and -MM that submitted their temperature and salinity on an isopycnic vertical grid; for these two models, we used the regularised z-level outputs.

We used only one ensemble member per model as even by the latest date of download, the majority of models had provided only one. We acknowledge that some models are not fully independent as they share similar codes (Table 1), and did not want

65  to accentuate the bias towards one model family by using different ensemble sizes. For most models, the ensemble member we used is referred to as r1i1p1f1. It was not available for CNRM-CM6-1, CNRM-ESM2-1, MIROC-ES2L and UKESM1-0-LL, for which we used r1i1p1f2. Neither was available for HadGEM3-GC31-LL, for which we used r1i1p1f3.





Although we used the full historical run for robustness verifications, we present only the results for the period January 1985 to December 2014, for consistency with the observational products. Note that we neither detrended the CMIP6 historical run nor substracted the pre-industrial control run, again for consistency with observations (which feature the climate change trend). These observations are the full-depth ocean temperature and salinity climatologies from the World Ocean Atlas 2018 (Locarnini et al., 2018; Zweng et al., 2018, respectively), the annual mixed layer depth climatology of de Boyer Montégut et al. (2004), and the global bathymetry GEBCO (GEBCO Compilation Group, 2019).

## 2.2 Computations: deep water properties, transports and extents

To start with, when necessary, we computed the monthly mixed layer depth (MLD) of the CMIP6 models as per the CMIP6 procedures by first computing the monthly mean density $\sigma_\theta$ from their monthly practical salinity and potential temperature. The MLD is then detected as the depth where $\sigma_\theta$ differs from that at 10 m depth by more than 0.125 kg m$^{-3}$. Note that a different threshold of 0.03 kg m$^{-3}$ is used in the observational reference (de Boyer Montégut et al., 2004). We could then quantify deep water formation in the three sectors of the Southern Ocean (borders at 65°W, 50°E and 130°E, south of 50°S, orange contours on Fig. 1), in the North Atlantic subpolar gyre (SPG, 70°W to 20°W, 50°N to 66°N) and in the Nordic seas (GIN, 30°W to 20°E, 65°N to 80°N , orange contours on Fig. 2) by computing the deep mixed volume (DMV) of each region as in Brodeau and Koenigk (2016). That is, for each month and each region, we keep only those grid cells where the MLD exceeds a critical value and sum the product MLD x cell area. We work with the maximum value of each year. As in Brodeau and Koenigk (2016) and Koenigk et al. (2020), we use a critical value of 700 m in the Nordic seas as it is the depth of the sill that connects them to the rest of the world ocean, and 1000 m in the Labrador Sea. As in e.g. Heuzé et al. (2013); De Lavergne et al. (2014), we use 2000 m in all three Southern Ocean sectors.

We quantify biases in the models by computing the root mean square error (model minus reference) in temperature, salinity and density $\sigma_\theta$ at the sea floor grid cell. To do so, all models had to be interpolated onto the reference's grid. Note that we purposely keep $\sigma_\theta$ instead of $\sigma_4$ as $\sigma_\theta$ is the density used in the models' code to notably compute the MLD. For later calculations, we also compute the temperature and salinity of the water masses AABW and NADW by taking their average properties over a specific region. As we will show in section 3.1, the AABW formation region really differs from model to model; as such, instead of using a limited region as in Johnson (2008), we detect AABW as having the temperature minimum anywhere deeper than 2000 m and south of 50°S. For NADW, we produce two flavours: $\mathrm{NADW_{SPG}}$ as having the salinity maximum anywhere deeper than 1000 m in the small area of SPG defined by Johnson (2008, 55°W to 54°W, 53°N to 63°N, yellow box on Fig. 2); and $\mathrm{NADW_{GIN}}$, the salinity maximum anywhere deeper than 1000 m in the GIN sector defined above.

We not only investigate the properties of AABW and NADW by their formation region, but also their transport into the rest of the global ocean. For AABW, we hence compute each model's Southern Meridional Overturning Circutlation or SMOC, using the same method as Heuzé et al. (2015) for comparison purposes. That is, in each ocean basin, we first integrate the meridional velocity vo from the west coast to the east coast at 30°S, then we integrate this value from the sea floor to the surface. The SMOC then is the northward deep maximum. We use a similar method for the AMOC, computing it at 35°N instead for comparison with the CMIP6 results of Menary et al. (2020). After integration from coast to coast and from sea




floor to surface of the velocity vo, the AMOC is defined as the southward subsurface maximum. We could not directly use the meridional overturning circulation output "msftmz" as

- – it is provided by hardly a third of the models;

– it is in kg s$^{-1}$ instead of m$^3$ s$^{-1}$, requiring division by the density, which we only have the monthly mean of;

- – for most models, the Indian and Pacific oceans are provided as one joint region, so we could not have obtained the SMOC in each basin.

Having to interpolate the irregular model grids onto the sections instead of directly using the model output may have introduced some errors. But as the AMOC results of this manuscript and that of Menary et al. (2020) for the models and experiment we
have in common are similar, we are confident in our MOC values. Note that two models, GFDL-ESM4 and NorCPM1, did not provide vo, limiting our transport analysis to 33 CMIP6 models.

Finally, to investigate in CMIP6 the link found in Heuzé et al. (2015) between the SMOCs and the northward extent of AABW layer, we chose to re-create for CMIP6 models the Johnson (2008) maps of AABW and NADW volumes in the global ocean. However, using the same approach as Johnson (2008) whereby we would have to determine the characteristics of every
water mass in each basin for each model is beyond the scope of this paper. Instead, as below the core of NADW the global ocean (excluding the Arctic) is a mixture of NADW and AABW only, we determine at each depth the NADW and AABW contents from a conservative property $\chi$ using the mixture equation of Jenkins (1999):

$$NADW_{content} = \frac{\chi_{AABW} - \chi}{\chi_{AABW} - \chi_{NADW}},$$ (1)

and

$$AABW_{content} = \frac{\chi_{NADW} - \chi}{\chi_{NADW} - \chi_{AABW}}.$$ (2)

Here, as in Johnson (2008), we consider the practical salinity and potential temperature as conservative enough to be used for these calculations. We then take the 50% content depth as the border between the NADW and AABW layers, i.e. anything with more than 50% AABW or less than 50% NADW is in the AABW layer, and the AABW thickness is the difference between the depth of that border and sea floor. We finally take the median of all the combinations: temperature or salinity, NADW properties
from SPG or GIN, and AABW or NADW contents. For the NADW layer, we detect the NADW core as the maximum NADW content with an extra criterion that the maximum must be larger than 80% NADW. Tests with values ranging from 60 to 100% yield similar values (not shown). Then the so-called NADW thickness is the thickness from the depth of the core to the NADW-AABW boundary (or to sea floor if there is no AABW). By working with a mixture of two water masses only, we could not try and detect the top of the NADW layer. Note that traditional methods of using a fixed temperature and/or salinity for water
mass determination cannot be applied to potentially biased climate models. The northward extent of AABW in each basin is defined as the northernmost latitude of the uninterrupted contour of thickness = 2000 m that starts in the Southern Ocean. We do the same for the southward extent of NADW in the Atlantic Ocean. For this part of the analysis, we show ony the results for NADW that originated in SPG; NADW that originated in GIN seems to leave the Nordic Seas in no model (not shown).



## 3   Results

In this section, we first look at deep water formation and properties in the Southern Ocean, then deep water formation and properties in the North Atlantic. It is only in the last section that we analyse both water masses together, by determining their global transports and volumes. In this section, we talk only about CMIP6. The comparison with CMIP5 will come in section 4.

### 3.1   Southern Ocean bottom water characteristics in CMIP6 models

#### 3.1.1   Shelf overflow and open ocean deep convection in the Southern Ocean

The "real" Antarctic Bottom Water forms on the continental shelves of the Weddell and Ross Sea, and then flows into the deep basins (visible as the densest areas on Fig. 1). The CMIP6 models' bottom density bias on the shelves suggests that 19/35 models may form dense water on the shelf: ACCESS-CM2 (Weddell and Ross), ACCESS-ESM1-5 (Ross), CAMS-CSM1-0 (Ross), the four CESM2 (Ross), CanESM5 (Weddell and Ross), GFDL-ESM4 (Weddell and Ross), the three GISS (Ross mainly), HadGEM3-GC31-LL (Weddell and Ross), INM-CM5 (Weddell and Ross), IPSL-CM6A-LR (Weddell and Ross), the

two NorESM2 (Ross mainly), SAM0-UNICON (Ross mainly), and UKESM1-0-LL (Weddell and Ross). The other 16 models are too light (strong negative bias on Fig. 1). Mean biases over 30 years are not enough to determine whether the dense shelf water flows into the deep ocean; we instead created movies of the monthly bottom density over the entire historical run for these 19 models, of which two are available as video supplement. The movies let us split these 19 models into three groups:

– INM-CM5 and NorESM2-LM show overflowing from the Ross shelf to the deep basin; so does NorESM2-MM, as well
as in the Amery sector (see video).

– GFDL-ESM4, HadGEM3-GC31-LL, IPSL-CM6A-LR, SAM0-UNICON and UKESM1-0-LL may overflow in the Ross sector, but we would need higher temporal resolution data to be certain;

– the other 11 models occasionally have a plume of dense water leaving the shelf, but it is nowhere near as dense as the shelf water it originates from (see video example of ACCESS-CM2).

In summary, in no model is there any (obvious) shelf export in the Weddell Sea. INM-CM5 (terrain following, high horizontal resolution model) and the two NorESM2 (hybrid isopycnic models) are the only ones forming AABW accurately via shelf processes, in the Ross sector only. The other high resolution models are not dense enough on the shelf or not clearly exporting their shelf water. There is no clear link between shelf processes and resolution (horizontal or vertical), vertical grid type, or ocean model component.



**Figure 1.** Southern Ocean reference bottom density $\sigma_\theta$ (top left panel, top colorbar), and for each CMIP6 model, bottom density bias (model minus reference) averaged over 1985-2014. White number for each model is its RMSE over the entire Southern Ocean deeper than 1000 m. Thick black line indicates maximum mixed layer deeper than 2000 m. Thin grey line, the 2000 m isobath. Orange lines on the reference panel delimitate the Weddell (W), Amery (A) and Ross (R) sectors for the DMV calculation (see methods).



If up to 8 models may export dense water from the shelf, how do the other 27 models of our study form their AABW? Deep convection, so far observed only once in the real ocean, in response to the 1974-1976 Weddell Polynya (Killworth, 1983), is the next obvious process to investigate. Mixed layer depths exceeding 2000 m are most prevalent in the Weddell sector (black contours on Fig. 1, DMVs in Table 2, MLD in supp. table A1). Of our 35 models, 24 exhibit deep convection in the Weddell Sea, of which 19 do so for most years of our study period. Most of these models also have a too large, re-occurring too often

Weddell Polynya (Mohrmann et al., subm.), except for GFDL-CM4 and IPSL-CM6A-LR who may be convecting under sea ice cover, and the two MIROC who are ice-free (Mohrmann et al., subm.; Roach et al., 2020). In the Amery and Ross sectors, we need to distinguish between the models that have non zero DMV because of open ocean deep convection, and those with coastal polynyas. In the Amery sector, aside from MCM-UA-1-0 whose deep convection area is but a continuation of the Weddell one, we have 10/35 models with open ocean deep convection: the two CNRM, the two EC-Earth3, GISS-E2-1-H,

MIROC6, MPI-ESM-1-2-HAM, MPI-ESM1-2-LR, and the two NorESM2. In the Ross sector, we have 15 models with open ocean deep convection, or to be more exact, 14 in the Ross Sea and NESM3 in the Amundsen Sea. In the Amery and Ross sectors, there is no link anymore between DMVs and the polynya activity by Mohrmann et al. (subm.), suggesting that deep water formation occurs under (thin?) ice cover. There is however a strong correlation of +0.47 between DMV in the Ross sector and DMV in the Weddell sector, i.e., models that convect a lot do it in both sectors. Behrens et al. (2016) suggests that a strong

DMV is associated with a strong Antarctic Circumpolar Current (ACC), while Cabré et al. (2017) find that strong DMV would weaken the westerly winds, i.e. may weaken the ACC. Here the only relationship between DMV and ACC, using the values from Beadling et al. (2020), is in agreement with Cabré et al. (2017): the more deep convection in the Amery sector, the weaker the ACC (correlation of -0.54, significant at 95% level). We find no relationship with DMV in the Weddell or Ross sectors.

    Up to now, there are still 7 models that have no open ocean deep convection and no shelf overflow: the four CESM2, GISS-

E2-1-G and -G-CC, and NorCPM1. GISS-E2-1-G and -G-CC have non zero DMV when considering the entire historical run (1850-2014, not shown), with GISS-E2-1-G convecting once in the Weddell sector and once in the Ross sector, and GISS-E2-1-G-CC thrice in the Weddell sector. The four CESM2 models do not, but they have an overflow parameterisation that artificially moves dense water from the shelf to the deep basins (Briegleb and Large, 2010). A pipe sucks the dense water on the shelves and releases it in the deep basin without having to cascade. This is why we cannot detect it on the overflow movies.

For NorCPM1 though, the mystery remains. Maybe it formed its AABW before 1850. In conclusions, most models form their AABW by open ocean deep convection. Even the models that seem to represent shelf processes accurately exhibit open ocean deep convection. Somewhat surprisingly, the only relationship between the DMV and the climate sensitivities of Zelinka et al. (2020) is in the Weddell sector (correlation of -0.36): models that convect a lot there have a low sensitivity, which is to be expected as heat and $CO_2$ are sent to the deep ocean. What is surprising is that the relationship holds only in the Weddell

sector. Hence, the sensitivity might be more linked to polynya activity, which is linked to deep convection only in the Weddell sector.




**Table 2.** Median and maximum deep mixing volume (DMV, see methods) for the Southern Ocean sectors (orange contours on Fig. 1) for each CMIP6 model over 1985-2014. Values given in $10^{13}\text{m}^3$, which is approximately the DMV of a 1x1° grid cell with a 1000 m mixed layer. Number in brackets indicates how many years out of 30 is the DMV different from zero, i.e. the number of years with deep convection

| | Weddell | | | Amery | | | Ross | | |
|---|---|---|---|---|---|---|---|---|---|
| model | median | max | (nb years) | median | max | (nb years) | median | max | (nb years) |
| ACCESS-CM2 | 161 | 526 | (30) | 0 | 0 | (0) | 178 | 311 | (30) |
| ACCESS-ESM1-5 | 408 | 588 | (30) | 0 | 0 | (0) | 66 | 266 | (24) |
| BCC-CSM2-MR | 432 | 764 | (27) | 1 | 11 | (16) | 0 | 0 | (0) |
| BCC-ESM1 | 596 | 1108 | (30) | 0 | 0 | (0) | 0 | 11 | (3) |
| CAMS-CSM1-0 | 128 | 415 | (30) | 1 | 7 | (22) | 0 | 13 | (2) |
| CESM2 | 0 | 0 | (0) | 0 | 0 | (0) | 0 | 0 | (0) |
| CESM2-FV2 | 0 | 0 | (0) | 0 | 0 | (0) | 0 | 0 | (0) |
| CESM2-WACCM | 0 | 0 | (0) | 0 | 0 | (0) | 0 | 0 | (0) |
| CESM2-WACCM-FV2 | 0 | 0 | (0) | 0 | 0 | (0) | 0 | 0 | (0) |
| CNRM-CM6-1 | 0 | 0 | (0) | 108 | 182 | (28) | 0 | 0 | (0) |
| CNRM-ESM2-1 | 0 | 0 | (0) | 67 | 300 | (30) | 0 | 0 | (0) |
| CanESM5 | 0 | 0.5 | (2) | 0 | 0 | (0) | 0 | 0 | (0) |
| EC-Earth3 | 0 | 403 | (12) | 0 | 86 | (11) | 0 | 0 | (0) |
| EC-Earth3-Veg | 0 | 212 | (12) | 17 | 78 | (20) | 0 | 0 | (0) |
| GFDL-CM4 | 1077 | 1334 | (30) | 0 | 6 | (9) | 16 | 53 | (30) |
| GFDL-ESM4 | 0 | 0 | (0) | 0 | 0 | (0) | 0 | 46 | (14) |
| GISS-E2-1-G | 0 | 0 | (0) | 0 | 0 | (0) | 0 | 0 | (0) |
| GISS-E2-1-G-CC | 0 | 0 | (0) | 0 | 0 | (0) | 0 | 0 | (0) |
| GISS-E2-1-H | 114 | 205 | (17) | 0 | 61 | (6) | 0 | 8 | (2) |
| HadGEM3-GC31-LL | 21 | 80 | (22) | 0 | 0 | (0) | 0 | 0.5 | (1) |
| INM-CM5-0 | 20 | 360 | (27) | 0 | 20 | (6) | 34 | 81 | (30) |
| IPSL-CM6A-LR | 56 | 168 | (30) | 0 | 0 | (0) | 0 | 2 | (6) |
| MCM-UA-1-0 | 11 | 13 | (30) | 0 | 1 | (26) | 3 | 5 | (30) |
| MIROC-ES2L | 169 | 581 | (27) | 0 | 0 | (0) | 280 | 581 | (30) |
| MIROC6 | 825 | 1117 | (30) | 29 | 80 | (28) | 930 | 1151 | (30) |
| MPI-ESM-1-2-HAM | 134 | 380 | (23) | 0 | 15 | (7) | 23 | 142 | (20) |
| MPI-ESM1-2-HR | 113 | 349 | (30) | 0 | 0 | (0) | 5 | 86 | (30) |
| MPI-ESM1-2-LR | 66 | 257 | (30) | 0 | 22 | (10) | 14 | 49 | (29) |
| MRI-ESM2-0 | 0 | 2 | (3) | 1 | 5 | (16) | 0 | 5 | (2) |
| NESM3 | 0 | 151 | (9) | 0 | 0 | (0) | 0 | 155 | (10) |
| NorCPM1 | 0 | 0 | (0) | 0 | 0 | (0) | 0 | 0 | (0) |
| NorESM2-LM | 678 | 873 | (30) | 9 | 50 | (19) | 0 | 24 | (12) |
| NorESM2-MM | 650 | 882 | (30) | 79 | 146 | (30) | 1 | 130 | (17) |
| SAM0-UNICON | 0 | 0 | (0) | 0 | 0 | (0) | 0 | 29 | (4) |
| UKESM1-0-LL | 1 | 69 | (17) | 0 | 0.4 | (2) | 0 | 0 | (0) |

### 3.1.2 AABW properties

Does the way CMIP6 models form their AABW impact its characteristics, as it did in CMIP5 (Heuzé et al., 2013)? Only density biases are shown on Fig. 1, but salinity and temperature biases are provided as supplementary Figs A1 and A2 respectively. Ten models have a negligible bottom density bias (RMSE lower than 0.05 kg m$^{-3}$, white numbers on Fig. 1): UKESM1-0-LL, CanESM5, IPSL-CM6A-LR, CESM2-WACCM-FV2, CESM2-FV2, CESM2, CESM2-WACCM, GISS-E2-1-H, CNRM-CM6 and HadGEM3-GC31-LL. Twelve more models have an acceptable bottom density bias (RMSE lower than 0.1 kg m$^{-3}$), including all the other models that are based on NEMO. That means that 22/35 models have acceptable biases. Let us investigate rather what may be common to the 13 models that are performing poorly.

INM-CM5 is the only model that is biased dense. Its bottom salinity is extremely high (RMSE of 0.42, supp. Fig. A1) while its bottom temperature is rather accurate (RMSE of 0.8°C, supp. Fig. A2). Its predecessor INM-CM4 had a similar issue, although whether this was caused by a too short spin-up or non-conservation of salt could not be determined (A. Gusev,





personal communication, July 2014). All the other 12 models are biased light (Fig. 1). For BCC-CSM2-MR, BCC-ESM1, MCM-UA-1-0, MIROC-ES2L, MIROC6 and MRI-ESM2-0, it is because of a fresh bias (supp. Fig. A1). The other 6 models

have relatively accurate bottom salinity, but are biased warm (supp. Fig. A2). No model has regional biases, which means that CMIP6 models are overall biased light or biased dense in the deep Southern Ocean (excluding the shelves).

The models with low biases in bottom density also tend to have zero to low DMVs in the Weddell Sea, but the relationship does not hold for maximum DMVs larger than $200 \ 10^{13} \ m^3$ (Table 2). NorESM2-LM and -MM notably have low biases but very high DMVs, but they also do shelf overflows. Open ocean deep convection leads to a warming and salinification of

bottom waters (Zanowski et al., 2015); one hypothesis is then that models that hardly convect stay closer to the bottom density value they were initialised with. In the case of the CESM2 suite, the overflow parameterisation may help form accurate bottom water. Biases as RMSE are not the whole story though. As expected, we do find significant relationship (95% level) between the actual temperature and salinity of AABW and the DMV: in the Amery and Ross sectors, more deep convection leads to warmer AABW (correlation of +0.33 and +0.29 respectively) as in Wang et al. (2017). In the Weddell sector however, more

deep convection leads to fresher AABW (correlation of -0.35), which in fact is consistent with the short-term response of the Southern Ocean to deep convection in Zanowski et al. (2015). The multimodel mean AABW salinity is 34.606 ± 0.154; the reference value from Johnson (2008), 34.641. The multimodel mean AABW temperature is -0.45 ± 0.73 °C; the reference value from Johnson (2008), -0.88°C. That is, the multimodel mean AABW is warmer and fresher than the reference, and more DMV worsens these biases. Note that the values of the individual models are given in supp. Table. A2.

To summarise, in the Southern Ocean, most models form their AABW by open ocean deep convection. In the Weddell Sea, this convection seems tied to the Weddell Polynya activity, and impacts the AABW salinity most: more deep convection, fresher bottom salinity. In the Amery and Ross sectors, it is linked more to the bottom temperature: more deep convection, warmer bottom salinity. Models which seem to form dense water via shelf processes also exhibit deep convection, so we cannot determine whether overflows alone would make the Southern Ocean more accurate. Models that convect the least or not

at all tend to be the most accurate; four or these, the CESM2 suite, may be aided by their overflow parameterisation (Briegleb and Large, 2010; Snow et al., 2015); another one, NorCPM1, assimilates observations (Counillon et al., 2016).

We will study the impact of these biases on the global transport of AABW in section 3.3. But as we cannot do so without investigating the AABW - NADW tug of war in the Atlantic basin, let us first evaluate the representation of NADW in CMIP6 models.

## 230 3.2 North Atlantic deep water in CMIP6 models

### 3.2.1 Deep water formation in the North Atlantic subpolar gyre and Nordic Seas

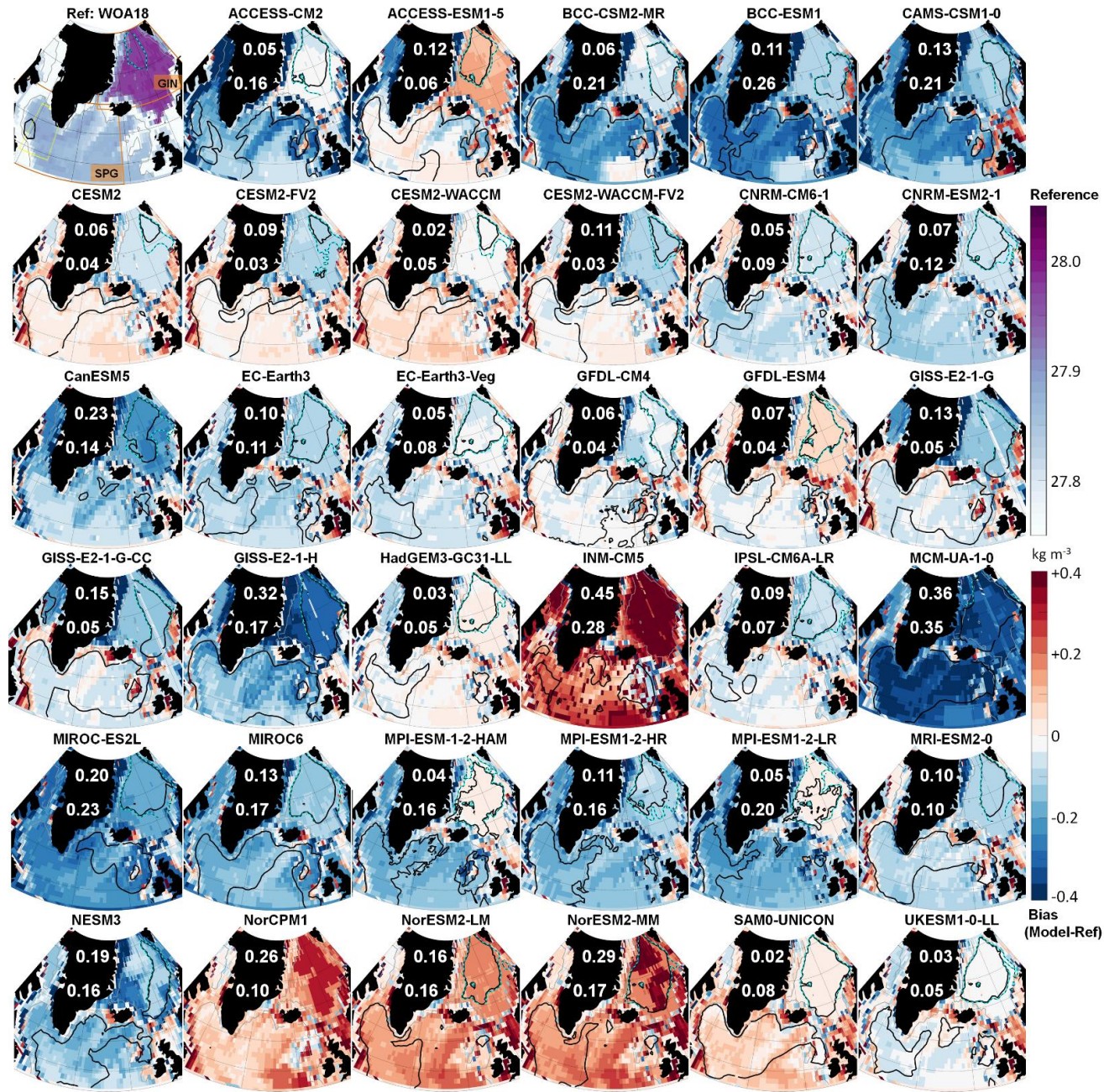

**Figure 2.** North Atlantic reference bottom density $\sigma_\theta$ (top left panel, top colorbar), and for each CMIP6 model, bottom density bias (model minus reference) averaged over 1985-2014. Orange lines on the reference panel delimitate the subpolar gyre (SPG) and Nordic Seas (GIN) areas for RMSE and DMV calculation; yellow dotted line, the SPG sector of Johnson (2008). White numbers for each model is its RMSE over the GIN (top) and SPG (bottom) areas, for depths over 1000 m. Thick black line indicates maximum mixed layer deeper than 1000 m; cyan dotted line in GIN, deeper than 700 m. Thin grey line, the 1000 m isobath.





In the North Atlantic, all 35 CMIP6 models of our study exhibit deep convection in the subpolar gyre (black contours on Fig. 2 and Table 3). As in CMIP5 (Heuzé, 2017), a large proportion of them convects not only in the Labrador Sea as the reference, but also intensely south of Iceland (Irminger Sea):

- 6/35 models convect only in the Labrador Sea (Fig. 2): CNRM-CM6-1, CNRM-ESM2-1, EC-Earth3-Veg, IPSL-CM6A-LR, NorCPM1 and NorESM2-LM;

- 9/35 models convect both in the Labrador and Irminger seas, but the two regions are not connected: ACCESS-CM2, BCC-ESM1, CESM2-WACCM-FV2, EC-Earth3, HadGEM3-GC31-LL, INM-CM5, MPI-ESM1-2-LR, NorESM2-MM and UKESM1-0-LL;

- 17/35 models convect both in the Labrador and Irminger seas, as one large SPG deep convection area: ACCESS-ESM1-5, BCC-CSM2-MR, CESM2, CESM2-FV2, CESM2-WACCM, GFDL-CM4, GFDL-ESM4, GISS-E2-1-G, GISS-E2-1-G-CC, GISS-E2-1-H, MCM-UA-1-0, MIROC6, MPI-ESM-1-2-HAM, MPI-ESM1-2-HR, MRI-ESM2-0, NESM3 and SAM0-UNICON;

- the last 3/35 models convect only in the Irminger Sea: CAMS-CSM1-0, CanESM5 and MIROC-ES2L.

As in Koenigk et al. (2020), the higher resolution NorESM2-MM and MPI-ESM1-2-HR have larger deep convection areas than their corresponding lower resolution NorESM2-LM and MPI-ESM1-2-LR. Note that the difference between NorESM2-MM and -LM is in the atmospheric component resolution. There is however no robust relationship across models between the horizontal resolution and the DMV. There is a relationship with the climate sensitivity though (Zelinka et al., 2020): the larger the DMV in SPG, the least sensitive the model (correlation of -0.36) which, as already discussed in the Southern Ocean part, 250 is not surprising. Consequently, there is also a strong correlation between the DMV in the Weddell Sea and in SPG (+0.57): models that convect a lot in the Weddell Sea convect a lot in SPG as well.

All models except INM-CM5-0 and NorCPM have deep convection in the GIN seas as well. Moreover, in GIN, models convect most years, with a minimum as high as 24/30 years (Table 3). There is more variability in the SPG, but likewise the majority of models convect all years. Besides, they convect too deep. While in the Southern Ocean, deep convection to the sea 255 floor can happen (Killworth, 1983), in the North Atlantic it should not go much beyond 1000 m (e.g. Våge et al., 2009). In the SPG, only the three models CanESM5, INM-CM5 and NorCPM1 have maximum mixed layer depths just exceeding 1000 m (supp table A1). An extra four models, ACCESS-CM2, CESM2, CESM2-WACCM and MIROC-ES2L, have tolerable depths up to 2500 m. But the vast majority convects too deep, often to the sea floor. It is the same in GIN, albeit with different models: this time, the four CESM2, MPI-ESM1-2-LR and NESM3 have MLDs up to 2000 m, and all the other models go to 3000 m or 260 even the sea floor. There is no significant correlation between the DMV in SPG and that in GIN. In summary, CMIP6 models exhibit deep convection in the North Atlantic too often, too deep, and over too large an area. It is not possible to determine the one most accurate model in the North Atlantic or even in each subregion; model users must choose a compromise between correct representation of the variability, location, depth or extent.





**Table 3.** Median and maximum deep mixing volume (DMV, see methods) for the subpolar gyre (SPG) and Nordic Seas (GIN, see Fig. 2) for each CMIP6 model over 1985-2014. Values given in $10^{13}$m$^3$, which is approximately the DMV of a 1x1° grid cell with a 1000 m mixed layer. Number in brackets indicates how many years out of 30 is the DMV different from zero, i.e. the number of years with deep convection

| model | SPG | | | GIN | | |
|---|---|---|---|---|---|---|
| | median | max | (nb years) | median | max | (nb years) |
| ACCESS-CM2 | 77 | 181 | (30) | 57 | 102 | (30) |
| ACCESS-ESM1-5 | 50 | 171 | (30) | 42 | 104 | (30) |
| BCC-CSM2-MR | 222 | 406 | (30) | 26 | 52 | (30) |
| BCC-ESM1 | 190 | 302 | (30) | 23 | 60 | (30) |
| CAMS-CSM1-0 | 107 | 225 | (30) | 12 | 35 | (24) |
| CESM2 | 73 | 238 | (30) | 6 | 24 | (30) |
| CESM2-FV2 | 118 | 206 | (30) | 6 | 15 | (30) |
| CESM2-WACCM | 76 | 226 | (30) | 8 | 23 | (30) |
| CESM2-WACCM-FV2 | 107 | 263 | (30) | 6 | 22 | (30) |
| CNRM-CM6-1 | 0 | 161 | (14) | 61 | 125 | (30) |
| CNRM-ESM2-1 | 3 | 78 | (25) | 51 | 134 | (30) |
| CanESM5 | 0 | 3 | (7) | 22 | 92 | (30) |
| EC-Earth3 | 20 | 141 | (26) | 60 | 103 | (30) |
| EC-Earth3-Veg | 18 | 201 | (20) | 65 | 103 | (30) |
| GFDL-CM4 | 417 | 548 | (30) | 36 | 70 | (30) |
| GFDL-ESM4 | 152 | 383 | (30) | 40 | 128 | (30) |
| GISS-E2-1-G | 303 | 476 | (30) | 32 | 103 | (30) |
| GISS-E2-1-G-CC | 316 | 418 | (30) | 29 | 86 | (29) |
| GISS-E2-1-H | 402 | 509 | (30) | 102 | 199 | (30) |
| HadGEM3-GC31-LL | 17 | 156 | (24) | 29 | 124 | (30) |
| INM-CM5-0 | 2 | 16 | (18) | 0 | 0 | (0) |
| IPSL-CM6A-LR | 0 | 43 | (12) | 47 | 109 | (30) |
| MCM-UA-1-0 | 2 | 3 | (30) | 1 | 1 | (30) |
| MIROC-ES2L | 1 | 56 | (19) | 122 | 208 | (30) |
| MIROC6 | 152 | 365 | (30) | 168 | 260 | (30) |
| MPI-ESM-1-2-HAM | 66 | 219 | (30) | 42 | 124 | (30) |
| MPI-ESM1-2-HR | 62 | 162 | (30) | 15 | 37 | (30) |
| MPI-ESM1-2-LR | 37 | 97 | (28) | 28 | 58 | (30) |
| MRI-ESM2-0 | 132 | 269 | (30) | 37 | 96 | (30) |
| NESM3 | 141 | 273 | (30) | 6 | 39 | (26) |
| NorCPM1 | 0 | 0 | (1) | 0 | 0 | (0) |
| NorESM2-LM | 150 | 278 | (30) | 34 | 97 | (30) |
| NorESM2-MM | 112 | 236 | (30) | 93 | 131 | (30) |
| SAM0-UNICON | 333 | 456 | (30) | 151 | 225 | (30) |
| UKESM1-0-LL | 25 | 129 | (20) | 109 | 177 | (30) |

### 3.2.2 North Atlantic bottom properties

The picture is less grim regarding bottom properties biases (shading on Fig. 2). Three models have bottom density biases resulting in an RMSE lower than 0.05 kg m$^{-3}$ in both SPG and GIN: CESM2-WACCM, HadGEM3-GC31-LL and UKESM1-0-LL. And extra 9 models have an RMSE lower than 0.1 kg m$^{-3}$ in both SPG and GIN: CESM2, CESM2-FV2, CNRM-CM6-1, EC-Earth3-Veg, GFDL-CM4, GFDL-ESM4, IPSL-CM6A-LR, MRI-ESM2-0 and SAM0-UNICON. As for the other 23 models, it depends on the region:

– INM-CM5, NorCPM1, NorESM2-M and NorESM2-MM are biased dense in both regions because they are biased salty (supp. Fig. A3). The magnitude of the bottom cell biases in INM-CM5 is very grid-cell dependent, maybe because of faulty regularisation of the sigma grid (even though it did not have this problem in the Southern Ocean).

– ACCESS-ESM1-5 is accurate in the SPG sector but biased dense (salty) in GIN.





- CESM2-WACCM-FV2, GISS-E2-1-G and GISS-E2-1-G-CC are accurate in the SPG sector but biased light in GIN. For
CESM2-WACCM-FV2, it is because of a warm bias (supp Fig. A4); for GISS-E2-1-G and GISS-E2-1-G-CC, because
of a fresh bias.

- ACCESS-CM2, BCC-CSM2-MR, CNRM-ESM2-1, EC-Earth3, MPI-ESM-1-2-HAM and MPI-ESM1-2-LR are biased
light in SPG but accurate in GIN. EC-Earth3 is the only one that is biased fresh. The other models are biased salty but
warm.

- The last 9 models are biased light in both regions. For CanESM5, GISS-E2-1-H, MCM-UA-1-0, MIROC-ES2L and
NESM3, this is caused by a salty bias; for BCC-ESM1, CAMS-CSM1-0, MIROC6 and MPI-ESM1-2-HR, a warm bias.

All models except CanESM5 and IPSL-CM6A-LR are in fact biased warm compared to the World Ocean Atlas 2018 bottom
temperature. The evolution of the bottom properties throughout the entire historical run is complex, with significantly different
variabilities depending on the model (not shown), and their analysis is beyond the scope of this paper. All that we can say that
the warm bias is not a result of only the modelled climate change or any drift.

In SPG, there is a strong relationship between the bottom temperature RMSE and the climate sensitivity of Zelinka et al.
(2020): the more sensitive the model, the least biased in temperature in SPG (correlation of -0.68). There is a somewhat signif-
icant (at the 90% level) relationship between the DMV and the bottom density bias in SPG only: the more the model convects,
the least biased (correlation of -0.29). There is however no relationship with the location of deep convection itself, e.g. MIROC-
ES2L that convects only in the Irminger Sea has a similar bias (magnitude and sign) as MPI-ESM1-2-LR that convects only in
the Labrador Sea; MPI-ESM1-LR in turns has a large bias and convects at the same location as UKESM1-0-LL, which has a
low bias. The four CESM2 models and their overflow parameterisation by Denmark Strait are again among the most accurate,
which was in fact the original motivation for that parameterisation (Briegleb and Large, 2010). NorCPM1 is somewhat disap-
pointing; it is built on NorESM2-LM and is supposed to have improved performances thanks to data assimilation (Counillon
et al., 2016). Its bottom density is indeed better than NorESM2-LM's in SPG, but is biased even denser (saltier) in GIN. There
is no across-model relationship between the sensitivity or the DMV and the salinity or GIN though, so the question for Nor-
CPM and other models remains. It may be linked to their respective biases in the representation of the Atlantic Multidecadal
Oscillation, as suggested by Lin et al. (2019). There is however a strong, convenient, interhemispheric correlation: the more
biased the bottom density in the Southern Ocean, the more biased it is in SPG (+0.81)

Finally, looking at the NADW properties instead of the biases (individual values in supp. table A2), we find that the multi-
model mean $NADW_{SPG}$, which Johnson (2008) refers to as upper NADW (UNADW) or Labrador Sea Water (LSW) is too
warm and too salty: $4.86 \pm 0.81°C$ and $35.163 \pm 0.143$ in CMIP6, instead of $3.32°C$ and $34.894$ in Johnson (2008). The
multimodel mean $NADW_{GIN}$ in contrast is accurate: $0.77 \pm 0.99°C$ and $35.001 \pm 0.169$ compared to $1.30°C$ and $34.878$
for the water mass called lower NADW (LNADW) or Iceland-Scotland Overflow Water (ISOW) in Johnson (2008), despite
5 models having a temperature below 0. Again in SPG, models with a high sensitivity have a lower salinity and temperature
(correlations of -0.31 and -0.45 respectively), which is consistent with the links previously found with the DMV. But again,
no relationship can be found in GIN. In GIN, we found no relationship between the biases or properties and the horizontal





or vertical resolution, nor the grid type or ocean model component. In CMIP5, Heuzé and Årthun (2019) had found strong across model biases in the inflow to the Nordic Seas caused by the large scale oceanic and atmospheric circulations as well as
the bathymetry, while Lin et al. (2019) showed that GIN property biases can be linked to the representation of multidecadal variability. Investigating the exact cause of the biases in GIN is beyond the scope of this paper, not least because in the next section, we will show that $NADW_{GIN}$ does not contribute to the global NADW in CMIP6 models. For now, we can conclude that the bottom property biases in GIN are not related to deep water formation in the region.

### 3.3 Global transport of NADW and AABW in CMIP6 models

In this last section, we shall determine the global fate of NADW and AABW once they leave their source regions. For NADW, this fate is tied to the strength of the Atlantic Meridional Overturning Circulation. The mean AMOC value lies at 18 Sv (1 Sv $= 10^6 \, \mathrm{m^3 \, s^{-1}}$), although observations both at the RAPID/MOCHA-array at 26.5°N (e.g. Duchez et al., 2016) and in the more recently deployed OSNAP-lines in the subpolar North Atlantic (Lozier et al., 2019), reveal a strong interannual variability of up to 5 Sv. Aside from INM-CM5 and its AMOC of $63 \pm 19$ Sv, all models fall in that range (individual values in supp. Table
B1), resulting in a multimodel median of $19.5 \pm 9.5$ Sv. Which means that like Menary et al. (2020), we find that the AMOC is overestimated in CMIP6 models on average, but not dramatically. Observations of the southern MOC at 30°S are rarer. From box inverse modelling, Lumpkin and Speer (2007) estimated the Atlantic SMOC at $5.6 \pm 3$ Sv; apart from the three GISS models and MIROC-ES2L that are too weak, all models are in that range (supp table B2), giving a multimodel median of $2.8 \pm 1.4$. Observational values in the Indian Ocean range between 3 and 27 Sv (Huussen et al., 2012), so unsurprisingly,
from the weak MCM-UA-1-0 ($1.5 \pm 1.6$ Sv) to the strong GFDL-CM4 ($11 \pm 18$ Sv), all models are in that range and the multimodel median is $3.0 \pm 2.5$ Sv. This is a remarkable improvement since CMIP5, where a majority of models had an Indian SMOC close to 0 (Heuzé et al., 2015). In the Pacific finally, Lumpkin and Speer (2007) estimated the MOC to be $11 \pm 5$ Sv. MCM-UA-1-0 is again the weakest ($3.9 \pm 1.9$ Sv), and the only model that falls out of the observational range, resulting in a multimodel median of $5.9 \pm 3.0$ Sv. In summary, the AMOC and southern MOCs are rather accurately represented in CMIP6
models!

The across-model correlations between the transports are strong and significant (95% level): the stronger the SMOC in the Indian Ocean, the stronger as well in the Pacific Ocean (correlation of +0.37). In contrast, a strong SMOC in either of these basins corresponds to a weak SMOC in the Atlantic (Atlantic-Indian, correlation of -0.45; Atlantic-Pacific, -0.34). And a weak SMOC in the Atlantic corresponds to a strong AMOC (correlation of -0.30), as previously found by Patara and Böning (2014)
in the NEMO model. We are obviously not implying causation from the correlations, but it is interesting to find relationships between the biases quantified in sections 3.1 and 3.2 and the transports. In agreement with Patara and Böning (2014), a stronger Atlantic SMOC is associated with lower temperature biases (correlation of 0.29), that is, colder AABW (-0.35), whereas a stronger Pacific SMOC is associated with stronger density biases (+0.36). A stronger AMOC is associated with larger biases in temperature and salinity in SPG (correlations of +0.33 and +0.37 respectively), and in particular a saltier $NADW_{SPG}$ (+0.34,
as in the paleoclimate simulations of Menviel et al., 2020). The Atlantic SMOC is the only transport that is linked to the strength the Antarctic Circumpolar Current (ACC, values from Beadling et al., 2020): the stronger the ACC, the stronger the





Atlantic SMOC (correlation of +0.37). There is no significant direct relationship between the transports and the DMV, which, at least for the AMOC, is in agreement with the recent observations of Lozier et al. (2019) and modelling work of Årthun et al. (2019). Unlike e.g. Menary et al. (2015) or Koenigk et al. (2020), we find no link between the MOCs and the models'
horizontal resolution.

In line with Heuzé et al. (2015), we expect the transports to impact the interbasin spread of NADW and AABW, that is, that the stronger the transport, the further from its source the water mass will travel. To investigate this, we recreated the AABW and NADW thickness maps of Johnson (2008) as Figs 3 and 4 respectively. For Fig. 4, we show only $NADW_{SPG}$; surprisingly, in no model could we find $NADW_{GIN}$ beyond the Nordic Seas (not shown). In agreement with observations and Johnson (2008),
AABW occupies the majority of the water column in most of the Indian and Pacific oceans, but its northward extent is limited in the Atlantic Ocean. Said extent is highly model dependent in the Atlantic, whereas it extends as far north as the basin limits permit in most models in the Indian and Pacific oceans. Finally, in most models, AABW in the Indian Ocean seems to come from the Pacific. The NADW southward expansion in the Atlantic is also model-dependent, with some reaching to the Antarctic Circumpolar Current (e.g. BCC-ESM1) and others not even leaving the North Atlantic subpolar gyre (e.g. UKESM1-0-LL).
As explained in the methods, the NADW layer in the Indian and Pacific oceans is most likely biased by our calculation method that takes into account only two water masses, and thus shall not be discussed further.

After extracting the southernmost extent of NADW and northernmost extents of AABW for each model (see supp. tables B1 and B2), we do find, as expected, that the stronger the AMOC, the further south NADW extends in the Atlantic (correlation of 0.32). And the stronger the Atlantic SMOC, the further north AABW extends in the Atlantic (correlation of 0.40). As we
previously found an anticorrelation between the AMOC and the Atlantic SMOC across CMIP6 models, the Atlantic balance is complete: models with strong AMOC and weak SMOC have their Atlantic dominated by NADW (e.g. CESM2-WACCM), whereas those with a weak AMOC and strong SMOC are filled with AABW (e.g. IPSL-CM6A-LR). And although there was no significant correlation between the DMV and the transports, we do find that the larger the DMV, the further the extent of NADW (DMV SPG, correlation of +0.34) or AABW (DMV Weddell, +0.51). We found no significant correlation between
the northward extent in the Indian or Pacific oceans and either the SMOCs or DMVs, or with the strength of the ACC. There are however relationships with their bottom properties: the northward extent of salty models is less than that of fresh models (correlations of -0.31 in the Indian and -0.44 in the Pacific). As we also find a strong positive relationship (correlation of +0.72) between the salinity of AABW and the salinity gradient across the ACC computed by Beadling et al. (2020), i.e., we find that the fresh models have a weak gradient to overcome, this result is not surprising. We can even speculate that in the absence of
NADW, AABW would expand further north in the fresher models regardless of their SMOC.



**Figure 3.** Thickness of the Antarctic Bottom Water layer in observations (top left panel) and in each CMIP6 models. See Methods.





**Figure 4.** Thickness of the North Atlantic Deep Water layer in observations (top left panel) and in each CMIP6 models, from the NADW core to its bottom. See Methods.





In conclusion, in CMIP6 models as in the real ocean, deep convection impacts bottom water characteristics and biases: in the Southern Ocean, deep convection seems associated with more biased deep waters; in the North Atlantic, the more the models convect, the least biased they are. Either way, these biases then impact the deep water transport: a saltier NADW is associated with a stronger AMOC; colder AABW, stronger Atlantic SMOC. These transports then impact the location of the "NADW -

AABW border" in the Atlantic: stronger AMOC and weaker Atlantic SMOC (the two transports are anticorrelated), further southward extent of NADW and less northward extent of AABW. In the Indian and Pacific oceans, the northward extent is larger in the fresher models, which are the ones with weak fronts in the ACC. To summarise, deep water formation is crucial for an accurate representation of the global deep ocean. We conclude this paper with a discussion of changes in deep water modelling since CMIP5, and what we can expect from the next generation(s) of simulations.

## 4 Discussion: changes since CMIP5 and way towards CMIP7

In CMIP5 models, no model assessed by Heuzé et al. (2013) could represent dense shelf overflows correctly. Consequently, models relied on open ocean deep convection for their deep water formation. The right amount of deep convection in the Weddell Sea was required for accurate bottom properties; models that convected too little or too much were the most biased. This relationship does not hold anymore for CMIP6, and it is the models that convect the least that tend to be the most accurate

(Fig. 1 and table 2). It may be because many models are now artificially prevented from opening polynyas and convecting in the Weddell Sea (Mohrmann et al., subm.). However, as the Weddell Polynya has now reopened in the real ocean (Campbell et al., 2019), future models may remove their "polynya-prevention" schemes again. Another reason for CMIP6 models seemingly not needing Southern Ocean deep convection to have accurate bottom properties may be that, as we showed in this paper, several CMIP6 models successfully represent shelf processes. This was an unexpected result considering that horizontal resolutions

have not increased much since CMIP5, suggesting that models have improved their parameterisations instead (Danek et al., 2019). Regardless of the formation process, bottom density biases are smaller in CMIP6 than they were in CMIP5 (RMSEs on Fig. 2 of Heuzé et al., 2013). The new version of the models that performed well in CMIP5 also performs well in CMIP6 (e.g. the IPSL and NorESM families), and the others have improved (CanESM4 had a bias of 0.17 kg m$^{-3}$; CanESM5, 0.03). The worst performing model of CMIP5 was INMCM4. The worst performing model of CMIP6 with respects to Southern Ocean

bottom properties is its successor, INM-CM5-0, but even this model saw its bias halve. INM-CM5-0 has both shelf processes and open ocean deep convection whereas INMCM4 had neither, which probably contributed to ridding the model of its cold bottom bias (Zanowski et al., 2015).

In the North Atlantic, to the best of our knowledge, most CMIP5 studies focussed on the relationship between deep water formation and the AMOC or the warming hole (e.g. Menary and Wood, 2018) but did not investigate bottom property biases.

The one exception is Ba et al. (2014) who found a recurrent cold bias; with the World Ocean Atlas 2018 as reference, we find in contrast that most CMIP6 models have a warm bias at the bottom of the North Atlantic. Deep water formation in the North Atlantic in the majority of CMIP5 models occurred too often, too deep, over too large an area (Heuzé, 2017). This sentence is still valid for CMIP6 (Fig. 2 and table 3). One noticeable improvement (?) is that the models whose CMIP5 predecessor



convected only in the Irminger Sea now convect in the entire subpolar gyre, including the Labrador Sea. Unfortunately, some
of the models that performed well in CMIP5 when considering the location of deep convection in the SPG, i.e. had a relatively
small area in the Labrador Sea, have also expanded to the entire SPG (e.g. the CNRM family). That is, the inaccurate models
may be on the way to improvement, most likely because the Arctic sea ice is better represented in CMIP6 than in CMIP5 (Shu
et al., 2020), but the ones that were relatively accurate have degraded. The same holds for the Nordic seas: CMIP6 models are
convecting even more than CMIP5 models did, and they already were convecting too much. In an increasingly warmer and ice-
free climate, Lique and Thomas (2018) predict that deep water formation would migrate from the North Atlantic subpolar gyre
to its subtropical gyre, and from the Nordic seas to the Arctic. Liu et al. (2019) adds that this will depend on whether meltwaters
will most strongly impact the stratification, shutting down deep convection, or the horizontal gradients and hence the winds,
pushing meltwater away from convection areas. For now, we observe that from the very icy CMIP5 to the more accurately
de-iced CMIP6 models, deep water formation regions just expanded to occupy most of the space available in SPG and GIN. It
is unclear whether increasing the resolution of future models would solve this issue: Danek et al. (2019) dramatically reduced
mixed layer depths in SPG by using an adaptative mesh with 5-15 km resolution, while Koenigk et al. (2020) finds that DMVs
in the SPG become even larger in the high resolution versions of the models that participated to HighResMIP. Without changing
the horizontal resolution, a more systematic inclusion and better representation of the stratosphere may be enough to reduce
deep convection in the North Atlantic (Haase et al., 2018).

Regarding the transports, as noted by Menary et al. (2020) the AMOC is stronger in CMIP6 than in CMIP5, which they
blame on the aerosol forcing. Except for INM-CM5 that is now way too strong, or which uploaded incorrect velocity fields,
this increase is not that strong and most models are in the obervational range. In the case of the CNRM family, a stronger
AMOC is in fact a much more accurate AMOC (from 12 Sv in CMIP5 to 19 Sv in CMIP6). The NorESM models have a
weaker AMOC in CMIP6, which is more accurate than their CMIP5 version (from 32 Sv in CMIP5 to 21 Sv in CMIP6). The
two highest resolution models have weakened so much that their AMOC is too low (GFDL-CM4 and MPI-ESM1-2-HR). This
seems in contradiction to Koenigk et al. (2020) who found that increased resolution in HighResMIP leads to a stronger AMOC,
but their result is mostly true when the models reach an eddy-resolving resolution. Which they do not, here, in CMIP6. It is
harder to determine whether the Southern MOCs at 30°S have improved since the values from inverse modelling (Lumpkin
and Speer, 2007) and observations (Huussen et al., 2012) have very large uncertainties. All that we can say is that the Atlantic
SMOC is stronger in CMIP6, so that only the GISS family continues having an Atlantic SMOC around 0 Sv. In the Indian
Ocean, no model has a transport of 0 anymore, which resulted in a doubling of the multimodel mean from 1.6 Sv in CMIP5
to 3 Sv in CMIP6, giving it the same importance as the Atlantic SMOC. The Pacific SMOC remains the strongest of the three
and sees no significant difference between CMIP5 and CMIP6 except for the two models that used to be around 0, INMCM4
(INM-CM5-0 is now at 10 Sv) and GISS-E2-H (GISS-E2-1-H now at 7 Sv). As in CMIP6 the Southern Ocean representation
from the bottom (this manuscript) to the top (Beadling et al., 2020) has improved, as well as the ACC (also Beadling et al.,
2020), it is no surprise that more models are now capable of exporting AABW to the rest of the world ocean. To the best of our
knowledge, the global extent of AABW and NADW, presented here for CMIP6 on Figs 3 and 4 respectively, was not assessed



in CMIP5, so we cannot determine whether improved Southern Ocean characteristics lead to an improved global water mass distribution.

What can we expect from a hypothetical CMIP7? Higher resolution, most likely, although that was already expected from CMIP6 and did not happen. As explained above and by Koenigk et al. (2020) or Danek et al. (2019), a higher resolution would not necessarily improve deep water formation. Holt et al. (2017) goes as far as stating that shelf processes will not be correctly represented until the horizontal resolution remains lower than 1/72°, which they expect might be reachable by the most advanced computers within 10 years. Unfortunately, we do not all have access to these computers, so that even now,

computing the global monthly mixed layer depth of the highest resolution model (GFDL-CM4, 1/4°) required over 600 core hours for the 165 years of the historical run. Higher resolution output will be impossible to manage, unless cloud-computing solutions such as PANGEO become the norm (Odaka et al., 2020). Instead of increasing the resolution, a seemingly easier solution would be to improve parameterisations (Holt et al., 2017), especially overflow parameterisations (Snow et al., 2015). Briegleb and Large (2010) first showed that an overflow parameterisation to transport water from the Nordic Seas to the rest

of the North Atlantic resulted in an improved representation of the ocean there. In CMIP6, the CESM2 models with their "pipes" in the North Atlantic and Antarctic shelves were among the most accurate models, especially for AABW. It would be interesting to see whether such a parameterisation on a different model would yield the same results, or whether the CESM2 models are just very accurate. Efforts could also concentrate on improving other components of the climate model, for example the atmosphere, as an improved representation of the stratosphere would supposedly decrease unrealistic deep water formation

(Haase et al., 2018). But where most progress can probably be made is in the cryosphere. As deep water formation is tied to the sea ice behaviour in both hemispheres, efforts such as sea ice MIP (SIMIP, Notz et al., 2016) dedicated to the modelling and coupling of sea ice may be the way forward. Likewise, the results of ice sheet MIP (ISMIP6, Nowicki et al., 2016) may shed a light on the debated impact of glacial meltwater on deep water formation (De Lavergne et al., 2014; Liu et al., 2019).

## 5   Conclusions

In this paper, we determined the characteristics of Antarctic Bottom Water and North Atlantic Deep Water in 35 models that participated in the latest installment of the Climate Model Intercomparison Project, CMIP6: their formation, properties, transports and extent in the global ocean. We focussed on the last thirty years of the historical run, January 1985 to December 2014. In the Southern Ocean (section 3.1), deep water formation is now more accurate, with several models representing shelf processes. Open ocean deep convection in the Weddell Polynya still happens in more than half of the models, but it is not a

requirement for accurate bottom water properties. In fact, the most accurate models were the ones with little to no open ocean convection, especially the CESM2 family that has an overflow parameterisation. In the North Atlantic (section 3.2), models convect too often, too deep, over too large an area, but in the subpolar gyre that area has migrated from the Irminger Sea (in CMIP5 models) to the more accurate Labrador Sea. The models that convect the most in the North Atlantic subpolar gyre also have the least biased NADW. NADW that forms in the subpolar gyre is the only one that occupies the world ocean; NADW

from the Nordic seas appears to stay in the Nordic seas. The saltier NADW, the stronger the AMOC, and the further south





the extent of NADW (section 3.3). That extent is limited by the strength of the abyssal overturning in the southern Atlantic or SMOC, with stronger Atlantic SMOC (caused by colder AABW) resulting in a further northward extent of AABW. In the Indian and Pacific oceans, the extent is directly related to the AABW properties, not the SMOCs: models with a comparatively fresh AABW are also the ones with weak fronts across the Antarctic Circumpolar Current, and hence can travel the furthest

north. In summary, for both deep water masses in CMIP6, their formation impacts their properties, which impact their transport and global extent, which in turns will have large impacts on global predictions of thermal expansion and sea level rise (Zickfeld et al., 2017), carbon storage (Tatebe et al., 2019), ecosystem changes (Sweetman et al., 2017) etc. Although CMIP6 models represent AABW and NADW more accurately than CMIP5 models did, a lot still need to be improved, especially deep water formation (section 4).

How to improve deep water formation in climate models then? A higher horizontal resolution may not be the answer as, depending on the model, it either reduces (Danek et al., 2019) or increases even further deep convection (Koenigk et al., 2020). In the ocean component, one solution could be a more systematic inclusion of overflow parameterisation (Snow et al., 2015); in this study, it seems very effective for CESM2. The one data-assimilating model, NorCPM (Counillon et al., 2016), also proposes an interesting option. In the rest of the model, improving the representation of the stratosphere seems effective at

reducing open ocean deep convection (Haase et al., 2018). Whatever the future holds, we hope it will feature a more systematic archiving of useful parameters. The situation has improved since CMIP5, but there are still CMIP6 models that do not provide their monthly mixed layer depth, and overturning streamfunctions (especially in density space) are a rarity. Making output directly available on cloud computing based systems such as PANGEO (Odaka et al., 2020) should also be a priority, to let researchers work on heavy CMIP data as soon as they are released, regardless of their computing and storage capacities.

*Code availability.* Codes can be provided upon reasonable request

*Data availability.* CMIP6 data are freely available via any portal of the Earth System Grid Federation; for this manuscript, we mostly used https://esgf-data.dkrz.de/projects/cmip6-dkrz/. The World Ocean Atlas 2018 data can be accessed freely at https://www.nodc.noaa.gov/OC5/ woa18/woa18data.html; the de Boyer Montégut et al. (2004) mixed layer depth reference data, at http://www.ifremer.fr/cerweb/deboyer/mld/ Surface_Mixed_Layer_Depth.php; the GEBCO reference bathymetry, at https://www.gebco.net/.

*Video supplement.* Two videos of monthly bottom density around Antarctica over the entire historical run are available as supplement: in ACCESS-CM2 that has no overflow (https://doi.org/10.5446/47545), and in NorESM2-MM, which exhibits overflows very clearly (https://doi.org/10.5446/47544)





## Appendix A: Bottom properties

In this appendix you will find:

- Figs A1 and A2, Southern Ocean bottom salinity and temperature respectively, to complement the bias discussion of section 3.1;

- Figs A3 and A4, North Atlantic bottom salinity and temperature respectively, to complement the bias discussion of section 3.2;

- Table A1, maximum mixed layer depth and convective area for each model and each region, corresponding to the DMVs
discussed in sections 3.1 and 3.2;

- Table A2, salinity and temperature of NADW and AABW for each model, briefly discussed in sections 3.1 and 3.2, and used for the thickness computations for Figs. 3 and 4.

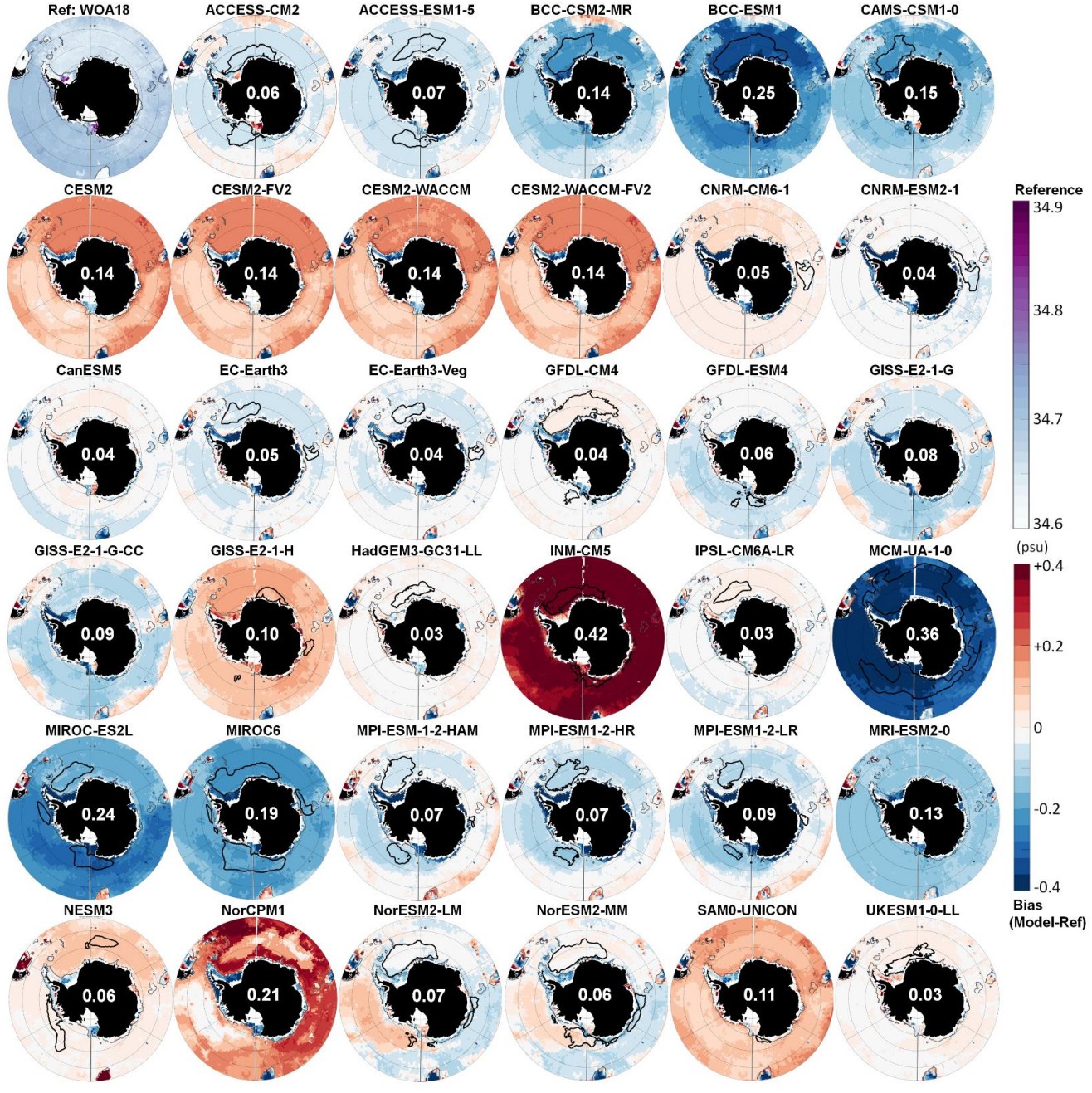

**Figure A1.** Southern Ocean reference bottom practical salinity (top left panel, top colorbar), and for each CMIP6 model, bottom practical salinity bias (model minus reference) averaged over 1985-2014. White number for each model is its RMSE over the entire Southern Ocean deeper than 1000 m. Thick black line indicates maximum mixed layer deeper than 2000 m. Thin grey line, the 2000 m isobath.


**Figure A2.** Southern Ocean reference bottom potential temperature (top left panel, top colorbar), and for each CMIP6 model, bottom potential temperature bias (model minus reference) averaged over 1985-2014. White number for each model is its RMSE over the entire Southern Ocean deeper than 1000 m. Thick black line indicates maximum mixed layer deeper than 2000 m. Thin grey line, the 2000 m isobath.





**Figure A3.** North Atlantic reference bottom practical salinity (top left panel, top colorbar), and for each CMIP6 model, bottom practical salinity bias (model minus reference) averaged over 1985-2014. White numbers for each model is its RMSE over the GIN (top) and SPG (bottom) areas, for depths over 1000 m. Thick black line indicates maximum mixed layer deeper than 1000 m; cyan dotted line in GIN, deeper than 700 m. Thin grey line, the 1000 m isobath.

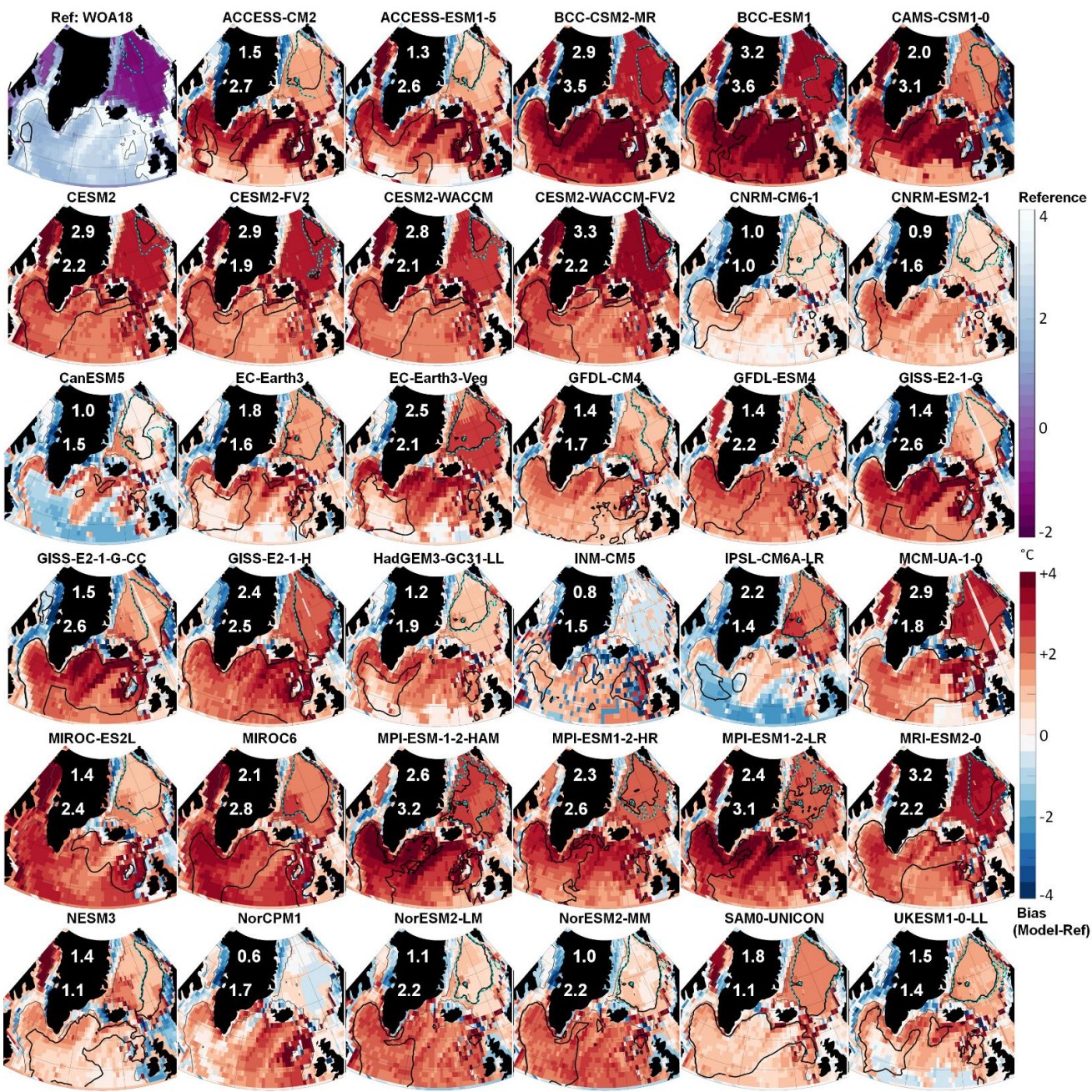

**Figure A4.** North Atlantic reference bottom potential temperature (top left panel, top colorbar), and for each CMIP6 model, bottom potential temperature bias (model minus reference) averaged over 1985-2014. White numbers for each model is its RMSE over the GIN (top) and SPG (bottom) areas, for depths over 1000 m. Thick black line indicates maximum mixed layer deeper than 1000 m; cyan dotted line in GIN, deeper than 700 m. Thin grey line, the 1000 m isobath.





**Table A1.** Supplementary table version of tables 2 and 3 showing 30 year max MLD (m) and max area (in $10\,000\ \mathrm{km}^2$, which is the approximate area of a $1°$ cell).

| model | SPG depth | SPG area | GIN depth | GIN area | Weddell depth | Weddell area | Amery depth | Amery area | Ross depth | Ross area |
|---|---|---|---|---|---|---|---|---|---|---|
| ACCESS-CM2 | 2550 | 123 | 3623 | 53 | 5087 | 134 | 0 | 0 | 4428 | 91 |
| ACCESS-ESM1-5 | 3013 | 108 | 3620 | 52 | 5328 | 125 | 0 | 0 | 4425 | 77 |
| BCC-CSM2-MR | 3787 | 216 | 3305 | 36 | 5334 | 183 | 3221 | 4 | 0 | 0 |
| BCC-ESM1 | 3787 | 160 | 3787 | 34 | 5334 | 317 | 0 | 0 | 3305 | 4 |
| CAMS-CSM1-0 | 2897 | 152 | 2810 | 31 | 5316 | 113 | 2698 | 3 | 3831 | 4 |
| CESM2 | 2280 | 143 | 2017 | 22 | 0 | 0 | 0 | 0 | 0 | 0 |
| CESM2-FV2 | 3102 | 115 | 2061 | 15 | 0 | 0 | 0 | 0 | 0 | 0 |
| CESM2-WACCM | 2392 | 130 | 2102 | 20 | 0 | 0 | 0 | 0 | 0 | 0 |
| CESM2-WACCM-FV2 | 2872 | 138 | 1858 | 22 | 0 | 0 | 0 | 0 | 0 | 0 |
| CNRM-CM6-1 | 3770 | 69 | 3699 | 59 | 0 | 0 | 4849 | 48 | 0 | 0 |
| CNRM-ESM2-1 | 3930 | 34 | 3699 | 59 | 0 | 0 | 4849 | 81 | 0 | 0 |
| CanESM5 | 1269 | 3 | 3216 | 69 | 2264 | 0.2 | 0 | 0 | 0 | 0 |
| EC-Earth3 | 3071 | 55 | 3699 | 64 | 5306 | 89 | 4496 | 28 | 0 | 0 |
| EC-Earth3-Veg | 4006 | 72 | 3699 | 58 | 5374 | 49 | 4737 | 23 | 0 | 0 |
| GFDL-CM4 | 4500 | 216 | 3500 | 36 | 6000 | 291 | 3500 | 2 | 4500 | 16 |
| GFDL-ESM4 | 3760 | 148 | 3734 | 60 | 0 | 0 | 0 | 0 | 4214 | 18 |
| GISS-E2-1-G | 4008 | 225 | 3342 | 48 | 0 | 0 | 0 | 0 | 0 | 0 |
| GISS-E2-1-G-CC | 4007 | 204 | 3341 | 42 | 0 | 0 | 0 | 0 | 0 | 0 |
| GISS-E2-1-H | 3000 | 263 | 3500 | 90 | 4500 | 63 | 3898 | 21 | 3159 | 3 |
| HadGEM3-GC31-LL | 3826 | 59 | 3699 | 53 | 5395 | 21 | 0 | 0 | 2333 | 0 |
| INM-CM5-0 | 1360 | 14 | 0 | 0 | 4500 | 111 | 2590 | 10 | 2886 | 37 |
| IPSL-CM6A-LR | 2686 | 20 | 3699 | 56 | 5036 | 60 | 0 | 0 | 2472 | 1 |
| MCM-UA-1-0 | 4662 | 1 | 3373 | 1 | 4662 | 3 | 4662 | 0.4 | 4662 | 1 |
| MIROC-ES2L | 2590 | 41 | 4065 | 88 | 6240 | 104 | 0 | 0 | 5190 | 156 |
| MIROC6 | 4740 | 144 | 4065 | 102 | 6240 | 234 | 5190 | 23 | 5190 | 289 |
| MPI-ESM-1-2-HAM | 3770 | 89 | 3395 | 70 | 5131 | 107 | 3760 | 4 | 4195 | 37 |
| MPI-ESM1-2-HR | 3388 | 101 | 3033 | 33 | 5170 | 94 | 0 | 0 | 4195 | 27 |
| MPI-ESM1-2-LR | 3395 | 47 | 1829 | 64 | 4872 | 68 | 3770 | 6 | 4195 | 18 |
| MRI-ESM2-0 | 4033 | 135 | 3650 | 52 | 2394 | 1 | 2958 | 2 | 2705 | 2 |
| NESM3 | 3292 | 118 | 1951 | 32 | 3772 | 50 | 0 | 0 | 4506 | 52 |
| NorCPM1 | 1005 | 0.3 | 0 | 0 | 0 | 0 | 0 | 0 | 0 | 0 |
| NorESM2-LM | 2741 | 133 | 3614 | 61 | 5410 | 215 | 4191 | 16 | 2844 | 10 |
| NorESM2-MM | 2770 | 130 | 2836 | 82 | 5408 | 217 | 4297 | 44 | 3323 | 51 |
| SAM0-UNICON | 3843 | 193 | 3380 | 98 | 0 | 0 | 0 | 0 | 2649 | 13 |
| UKESM1-0-LL | 3750 | 52 | 3699 | 85 | 5037 | 22 | 2195 | 0.2 | 0 | 0 |





**Table A2.** For each CMIP6 model, 30 year median practical salinity S and potential temperature $\theta$ (°C) of the NADW formed in the subpolar gyre (SPG) or Nordic seas (GIN), and of the AABW.

| model | NADW (SPG) S | $\theta$ | NADW (GIN) S | $\theta$ | AABW S | $\theta$ |
|---|---|---|---|---|---|---|
| ACCESS-CM2 | 35.126 ± 0.024 | 5.95 ± 0.26 | 34.988 ± 0.021 | 0.65 ± 0.17 | 34.613 ± 0.045 | -0.84 ± 0.15 |
| ACCESS-ESM1-5 | 35.328 ± 0.050 | 5.89 ± 0.32 | 35.176 ± 0.031 | 0.46 ± 0.26 | 34.581 ± 0.005 | -1.06 ± 0.02 |
| BCC-CSM2-MR | 35.155 ± 0.038 | 6.00 ± 0.19 | 35.129 ± 0.002 | 1.92 ± 0.00 | 34.440 ± 0.025 | -0.44 ± 0.05 |
| BCC-ESM1 | 35.086 ± 0.022 | 6.12 ± 0.16 | 35.043 ± 0.010 | 2.20 ± 0.17 | 34.308 ± 0.011 | -0.91 ± 0.09 |
| CAMS-CSM1-0 | 35.104 ± 0.004 | 6.04 ± 0.14 | 34.909 ± 0.001 | 0.77 ± 0.01 | 34.423 ± 0.011 | -0.28 ± 0.07 |
| CESM2 | 35.264 ± 0.018 | 5.37 ± 0.18 | 35.064 ± 0.010 | 1.97 ± 0.11 | 34.677 ± 0.030 | -0.50 ± 0.34 |
| CESM2-FV2 | 35.242 ± 0.027 | 5.17 ± 0.22 | 35.023 ± 0.005 | 1.84 ± 0.10 | 34.713 ± 0.014 | -0.63 ± 0.08 |
| CESM2-WACCM | 35.260 ± 0.025 | 5.22 ± 0.19 | 35.112 ± 0.004 | 1.75 ± 0.07 | 34.682 ± 0.016 | -0.31 ± 0.15 |
| CESM2-WACCM-FV2 | 35.258 ± 0.026 | 5.40 ± 0.19 | 35.036 ± 0.013 | 2.39 ± 0.07 | 34.683 ± 0.017 | -0.53 ± 0.10 |
| CNRM-CM6-1 | 34.968 ± 0.030 | 3.57 ± 0.19 | 34.927 ± 0.025 | 0.06 ± 0.33 | 34.679 ± 0.001 | 0.25 ± 0.02 |
| CNRM-ESM2-1 | 35.009 ± 0.026 | 4.12 ± 0.13 | 34.904 ± 0.024 | 0.12 ± 0.26 | 34.641 ± 0.003 | 1.00 ± 0.04 |
| CanESM5 | 34.981 ± 0.008 | 4.77 ± 0.07 | 34.799 ± 0.017 | 0.54 ± 0.10 | 34.655 ± 0.015 | -1.07 ± 0.02 |
| EC-Earth3 | 35.086 ± 0.028 | 4.90 ± 0.13 | 34.902 ± 0.060 | 0.46 ± 0.64 | 34.601 ± 0.000 | 0.13 ± 0.01 |
| EC-Earth3-Veg | 35.261 ± 0.018 | 5.56 ± 0.12 | 35.068 ± 0.024 | 1.73 ± 0.23 | 34.607 ± 0.014 | 0.17 ± 0.02 |
| GFDL-CM4 | 35.179 ± 0.025 | 5.25 ± 0.21 | 34.936 ± 0.004 | 1.20 ± 0.63 | 34.460 ± 0.019 | -1.08 ± 0.10 |
| GFDL-ESM4 | 35.251 ± 0.021 | 5.40 ± 0.16 | 35.102 ± 0.017 | 0.58 ± 0.17 | 34.563 ± 0.037 | -0.57 ± 0.15 |
| GISS-E2-1-G | 35.197 ± 0.027 | 4.81 ± 0.14 | 34.843 ± 0.004 | 0.39 ± 0.05 | 34.559 ± 0.011 | -0.31 ± 0.08 |
| GISS-E2-1-G-CC | 35.193 ± 0.039 | 4.80 ± 0.18 | 34.840 ± 0.006 | 0.57 ± 0.05 | 34.551 ± 0.010 | -0.23 ± 0.08 |
| GISS-E2-1-H | 35.010 ± 0.024 | 3.96 ± 0.31 | 34.719 ± 0.008 | 1.47 ± 0.15 | 34.788 ± 0.033 | 0.96 ± 0.02 |
| HadGEM3-GC31-LL | 35.193 ± 0.013 | 4.72 ± 0.09 | 34.991 ± 0.008 | 0.07 ± 0.03 | 34.628 ± 0.032 | -0.17 ± 0.16 |
| INM-CM5-0 | 35.310 ± 0.007 | 2.97 ± 0.21 | 35.512 ± 0.003 | -1.59 ± 0.01 | 35.106 ± 0.017 | -0.45 ± 0.12 |
| IPSL-CM6A-LR | 35.032 ± 0.018 | 3.89 ± 0.11 | 35.001 ± 0.005 | 1.70 ± 0.05 | 34.662 ± 0.007 | -1.25 ± 0.01 |
| MCM-UA-1-0 | 34.697 ± 0.039 | 4.10 ± 0.31 | 34.708 ± 0.009 | 1.60 ± 0.29 | 34.265 ± 0.003 | -1.23 ± 0.10 |
| MIROC-ES2L | 34.946 ± 0.007 | 5.25 ± 0.25 | 34.781 ± 0.014 | 0.73 ± 0.10 | 34.347 ± 0.032 | 1.56 ± 0.20 |
| MIROC6 | 35.081 ± 0.004 | 4.86 ± 0.03 | 34.953 ± 0.015 | 1.55 ± 0.07 | 34.419 ± 0.018 | 0.22 ± 0.13 |
| MPI-ESM-1-2-HAM | 35.195 ± 0.013 | 5.94 ± 0.13 | 35.128 ± 0.003 | 1.50 ± 0.04 | 34.581 ± 0.012 | -0.12 ± 0.05 |
| MPI-ESM1-2-HR | 35.175 ± 0.033 | 6.07 ± 0.22 | 34.954 ± 0.003 | 1.05 ± 0.06 | 34.554 ± 0.002 | -0.35 ± 0.04 |
| MPI-ESM1-2-LR | 35.163 ± 0.017 | 6.10 ± 0.18 | 35.101 ± 0.001 | 1.19 ± 0.02 | 34.554 ± 0.009 | -0.01 ± 0.05 |
| MRI-ESM2-0 | 35.094 ± 0.024 | 4.22 ± 0.33 | 35.042 ± 0.009 | 2.16 ± 0.11 | 34.526 ± 0.025 | -1.21 ± 0.10 |
| NESM3 | 34.894 ± 0.013 | 3.96 ± 0.65 | 34.896 ± 0.001 | -0.36 ± 0.00 | 34.689 ± 0.000 | 0.84 ± 0.01 |
| NorCPM1 | 35.227 ± 0.011 | 4.49 ± 0.08 | 35.276 ± 0.000 | -1.52 ± 0.00 | 34.659 ± 0.000 | -0.72 ± 0.00 |
| NorESM2-LM | 35.373 ± 0.013 | 4.54 ± 0.13 | 35.157 ± 0.024 | -0.31 ± 0.16 | 34.543 ± 0.115 | -1.48 ± 0.11 |
| NorESM2-MM | 35.391 ± 0.014 | 4.48 ± 0.21 | 35.376 ± 0.042 | -0.74 ± 0.44 | 34.796 ± 0.137 | -1.57 ± 0.07 |
| SAM0-UNICON | 35.138 ± 0.030 | 4.43 ± 0.66 | 35.055 ± 0.007 | 0.93 ± 0.06 | 34.745 ± 0.016 | -1.04 ± 0.07 |
| UKESM1-0-LL | 35.141 ± 0.041 | 4.43 ± 0.21 | 34.990 ± 0.026 | 0.38 ± 0.22 | 34.639 ± 0.037 | -0.74 ± 0.20 |
| multimodel mean | 35.163 ± 0.143 | 4.86 ± 0.81 | 35.001 ± 0.169 | 0.77 ± 0.99 | 34.607 ± 0.154 | -0.45 ± 0.73 |

## Appendix B: Transports

In this section, you will find two tables to complement section 3.3:

- Table B1 presents the AMOC and southernmost extent of NADW in each model;

- Table B2 presents the SMOC and northernmost extent of AABW in the Atlantic, Indian and Pacific oceans, in each model.





**Table B1.** For each CMIP6 model, 30 year median AMOC at 35°N (in Sv), and southernmost latitude (in degrees north) of the 2000 m -thick NADW layer in the Atlantic from Fig. 4.

| model | AMOC | latitude |
|---|---|---|
| ACCESS-CM2 | 19.8 ± 3.5 | 48.5 |
| ACCESS-ESM1-5 | 20.0 ± 2.1 | 44.5 |
| BCC-CSM2-MR | 26.1 ± 2.8 | -49.5 |
| BCC-ESM1 | 25.3 ± 3.7 | -59.5 |
| CAMS-CSM1-0 | 14.8 ± 3.5 | 50.5 |
| CESM2 | 24.9 ± 6.5 | -49.5 |
| CESM2-FV2 | 25.6 ± 6.4 | -50.5 |
| CESM2-WACCM | 24.7 ± 7.3 | -49.5 |
| CESM2-WACCM-FV2 | 24.8 ± 7.0 | -50.5 |
| CNRM-CM6-1 | 19.4 ± 4.9 | -46.5 |
| CNRM-ESM2-1 | 20.0 ± 5.1 | -47.5 |
| CanESM5 | 15.1 ± 5.7 | 60 |
| EC-Earth3 | 17.8 ± 5.1 | 41.5 |
| EC-Earth3-Veg | 19.5 ± 5.0 | 31.5 |
| GFDL-CM4 | 8.9 ± 10.7 | -43.5 |
| GFDL-ESM4 | N/A | -48.5 |
| GISS-E2-1-G | 22.2 ± 4.8 | -51.5 |
| GISS-E2-1-G-CC | 24.0 ± 4.1 | -50.5 |
| GISS-E2-1-H | 16.4 ± 12.9 | 60 |
| HadGEM3-GC31-LL | 18.8 ± 4.0 | 50.5 |
| INM-CM5-0 | 63.1 ± 19.8 | 60 |
| IPSL-CM6A-LR | 13.4 ± 5.3 | 60 |
| MCM-UA-1-0 | 17.9 ± 2.4 | 8.5 |
| MIROC-ES2L | 15.0 ± 6.0 | 47.5 |
| MIROC6 | 19.0 ± 6.2 | 47.5 |
| MPI-ESM-1-2-HAM | 29.6 ± 8.7 | 2.5 |
| MPI-ESM1-2-HR | 1.8 ± 12.8 | -50.5 |
| MPI-ESM1-2-LR | 25.4 ± 7.4 | -13.5 |
| MRI-ESM2-0 | 18.6 ± 17.7 | -48.5 |
| NESM3 | 8.8 ± 4.2 | 19.5 |
| NorCPM1 | N/A | -49.5 |
| NorESM2-LM | 18.0 ± 7.3 | 60 |
| NorESM2-MM | 21.4 ± 7.6 | 40.5 |
| SAM0-UNICON | 24.9 ± 5.6 | -50.5 |
| UKESM1-0-LL | 18.6 ± 4.5 | 40.5 |
| multimodel median | 19.5 ± 9.5 | |





**Table B2.** For each CMIP6 model, 30 year median southern MOC at 30°S (SMOC, in Sv) and northernmost latitude (in degrees north) of the 2000 m -thick AABW layer in each ocean from Fig. 3.

| model | Atlantic SMOC | lat. | Indian SMOC | lat. | Pacific SMOC | lat. |
|---|---|---|---|---|---|---|
| ACCESS-CM2 | 4.2 ± 1.2 | -35.5 | 1.5 ± 3.1 | 16.5 | 3.9 ± 3.8 | 54.5 |
| ACCESS-ESM1-5 | 3.6 ± 1.0 | -34.5 | 1.9 ± 3.3 | 21.5 | 5.9 ± 3.5 | 54.5 |
| BCC-CSM2-MR | 2.7 ± 1.1 | 7.5 | 4.8 ± 9.1 | 16.5 | 6.7 ± 3.5 | 57.5 |
| BCC-ESM1 | 4.2 ± 1.5 | 13.5 | 3.6 ± 8.3 | 17.5 | 8.2 ± 2.7 | 58.5 |
| CAMS-CSM1-0 | 5.9 ± 1.8 | 8.5 | 1.6 ± 4.6 | -12.5 | 2.8 ± 3.5 | 58.5 |
| CESM2 | 2.3 ± 1.2 | -38.5 | 2.2 ± 2.8 | -49.5 | 3.9 ± 4.0 | 52.5 |
| CESM2-FV2 | 2.3 ± 1.2 | -40.5 | 2.0 ± 3.1 | -51.5 | 3.5 ± 3.7 | 17.5 |
| CESM2-WACCM | 2.1 ± 1.2 | -38.5 | 1.9 ± 3.0 | -39.5 | 3.3 ± 3.7 | 53.5 |
| CESM2-WACCM-FV2 | 2.7 ± 1.3 | -38.5 | 1.6 ± 3.0 | -50.5 | 3.2 ± 3.7 | 52.5 |
| CNRM-CM6-1 | 1.5 ± 1.7 | -26.5 | 3.4 ± 4.6 | 21.5 | 6.3 ± 4.8 | 57.5 |
| CNRM-ESM2-1 | 1.8 ± 1.6 | -31.5 | 3.1 ± 4.8 | 19.5 | 6.2 ± 4.5 | 60.5 |
| CanESM5 | 4.0 ± 1.6 | -3.5 | 3.7 ± 2.8 | 11.5 | 6.3 ± 3.4 | 57.5 |
| EC-Earth3 | 3.8 ± 2.2 | 14.5 | 1.5 ± 4.5 | 14.5 | 4.9 ± 4.2 | 60.5 |
| EC-Earth3-Veg | 2.8 ± 2.1 | -25.5 | 1.3 ± 4.8 | 25.5 | 4.3 ± 4.1 | 59.5 |
| GFDL-CM4 | 3.0 ± 2.4 | 13.5 | 11.1 ± 18.2 | 11.5 | 3.2 ± 3.1 | 59.5 |
| GFDL-ESM4 | N/A | 14.5 | N/A | 25.5 | N/A | 60.5 |
| GISS-E2-1-G | 0.4 ± 0.5 | -40.5 | 8.7 ± 5.7 | -43.5 | 10.1 ± 5.8 | -42.5 |
| GISS-E2-1-G-CC | 0.3 ± 0.5 | -41.5 | 8.9 ± 5.6 | -37.5 | 10.9 ± 6.2 | -44.5 |
| GISS-E2-1-H | 0.2 ± 1.6 | -33.5 | 10.3 ± 8.3 | -40.5 | 7.3 ± 6.7 | -53.5 |
| HadGEM3-GC31-LL | 3.1 ± 1.9 | -26.5 | 2.3 ± 3.4 | 25.5 | 7.1 ± 4.8 | 59.5 |
| INM-CM5-0 | 3.4 ± 1.9 | 52.5 | 3.0 ± 2.8 | -50 | 10.8 ± 3.9 | -50 |
| IPSL-CM6A-LR | 3.8 ± 2.6 | -2.5 | 2.3 ± 4.4 | 16.5 | 5.8 ± 5.4 | 60.5 |
| MCM-UA-1-0 | 3.5 ± 0.6 | -33.5 | 1.5 ± 1.6 | 21.5 | 3.9 ± 1.9 | -23.5 |
| MIROC-ES2L | 0.3 ± 0.5 | 13.5 | 5.1 ± 4.2 | 17.5 | 12.1 ± 5.3 | 58.5 |
| MIROC6 | 4.0 ± 1.3 | 14.5 | 5.1 ± 3.7 | 23.5 | 13.6 ± 4.0 | 60.5 |
| MPI-ESM-1-2-HAM | 2.9 ± 1.5 | 13.5 | 3.1 ± 5.3 | -30.5 | 3.0 ± 3.4 | 60.5 |
| MPI-ESM1-2-HR | 5.8 ± 2.1 | 13.5 | 4.1 ± 4.9 | -17.5 | 5.1 ± 3.5 | 60.5 |
| MPI-ESM1-2-LR | 2.9 ± 1.6 | 13.5 | 3.3 ± 4.9 | -30.5 | 2.9 ± 3.7 | 60.5 |
| MRI-ESM2-0 | 2.8 ± 1.2 | -27.5 | 2.6 ± 4.0 | 25.5 | 6.6 ± 5.6 | 56.5 |
| NESM3 | 1.4 ± 1.4 | -6.5 | 2.3 ± 4.8 | -4.5 | 4.4 ± 4.7 | 57.5 |
| NorCPM1 | N/A | -47.5 | N/A | -47.5 | N/A | -23.5 |
| NorESM2-LM | 1.7 ± 1.1 | -39.5 | 3.3 ± 4.3 | -28.5 | 10.5 ± 4.8 | 60.5 |
| NorESM2-MM | 1.4 ± 1.0 | 5.5 | 4.0 ± 4.8 | 14.5 | 10.6 ± 5.3 | 59.5 |
| SAM0-UNICON | 1.7 ± 1.1 | -42.5 | 2.9 ± 3.8 | -49.5 | 5.6 ± 4.5 | -11.5 |
| UKESM1-0-LL | 3.4 ± 1.9 | -36.5 | 3.0 ± 3.3 | 25.5 | 9.0 ± 4.2 | 59.5 |
| multimodel median | 2.8 ± 1.4 | | 3.0 ± 2.5 | | 5.9 ± 3.0 | |



*Author contributions.* C.H. designed and conducted the analyses, and wrote the paper.

*Competing interests.* The author declares no competing interest.

*Acknowledgements.* This work is supported by the Swedish Research Council (dnr. 2018-03859). We acknowledge the World Climate Research Programme's Working Group on Coupled Modelling, which is responsible for CMIP, and we thank the climate modeling groups (whose models are listed in Table 1 of this paper) for producing and making available their model output. C.H. would like to thank Jonathan Rheinlænder for the constructive discussion that inspired this manuscript, Martin Mohrmann for the regular CMIP6 deep convection chats during its writing, and Matthew Menary for freely sharing his AMOC data (in fact, not used in this manuscript).





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
