# Peer review of "Antarctic Bottom Water and North Atlantic Deep Water in CMIP6 models"

_Ocean Science, 2020_

## Referee Comment (RC1) · Anonymous Referee #1 · 25 Jul 2020

This manuscript discusses differences of ocean water properties between observations and the CMIP6 suite of coupled climate models, focusing on density, temperature, and salinity at the sea floor in the North Atlantic and Southern Ocean, where North Atlantic Deep Waters and Antarctic Bottom Waters are formed, as well as thicknesses of both of those water masses. The work is interesting and original. The manuscript should be worthy of publication after revision. However, there are some inaccurate statements in the manuscript that should be corrected before acceptance. Furthermore, one major suggestion is offered, which would hopefully increase the usefulness of the work substantially, but would require some more text. That one major suggestion, that the author might consider, then a detailed list of comments, follow.

Major suggestion:

[Figure]

1. While the 6x6 postage stamp figure format is quite useful, especially for the modelers, it may be a bit much for some readers. Please consider putting all the postage stamp figures into the appendix and switching the text figures to 3-panel figures (water property reference, mean or median difference of water property from reference, std deviation or interquartile range of water property difference from reference). This would keep 4 figures in the main text:

Two 3x3 figures (3 panels each of reference, mean, and standard deviation for potential temperature, practical salinity, and sigma-4) in the S. Ocean and N. Atlantic.

Two 1x3 figures of reference, mean differences, and standard deviations of AABW thickness and NADW thickness

This change would mean eight postage stamp "difference" figures in the appendix (NADW thickness, AABW thickness, and temper, salinity, and sigma-4 for both the Southern Ocean and North Atlantic.

This suggested change would also require some additional text to describe the means and standard deviations of the differences of the models from observations, but it might be worth it. The statistical figures would probably be much more widely used than the postage stamp figures. They would be great in a general presentation on climate model evaluation.

Major comments:

2. L1. The first sentence is arguably inaccurate, as the global ocean circulation is mostly wind driven. Maybe "Deep and bottom water formation are an important part of the global ocean circulation" would be better.

3. Throughout. Please use the commonly accepted terminology "bottom water" to distinguish Antarctic Bottom Water from "deep water" (North Atlantic Deep Water). The manuscript may be slightly longer as a result (as sometimes the use of "deep and bottom water" will be required), but will be more in line with the field, and easier to

read.

4. L19. The opening sentence of the introduction doesn't seem accurate, and it is difficult to parse the meaing. How about "Bottom water formation around Antarctica and deep water formation in the North Atlantic ventilate the global abyssal and deep ocean."?

5. L23. This sentence does not seem quite right, as AABW influence spreads quite far into the N. Atlantic, albeit in a form highly diluted by mixing with NADW above. How about "In a substantial portion of the Atlantic, Antarctic Bottom Water spreading north is overlain by North Atlantic Deep Water spreading south."

6. L63-65. This sentence conflates two different issues, and perhaps should be broken in two at the "and".

7. L140-142. This portion is overly concise. How about something along the lines of: "Presently, Antarctic Bottom water is primarily formed in several locations (including the Weddell Sea, the Ross Sea, and the Adelie Lands) as water is cooled, made saltier, and denser on the continental shelves, then cascades down the continental slopes, entraining deep waters on its way to the sea floor."

8. L160-161. The recent Polynya in the Weddell Sea that is mentioned later in the paper should also be mentioned here.

9. Tables 2 and 3. Consider that instead of years out of 30, a percentage would be more quickly and easily interpreted by most readers.

10. L205. How is it possible that "No model has regional biases"? Perhaps just delete the part of this sentence on L206, and start the sentence with "CMIP6 models tend to be either biased light or biased dense".

11. L265. Consider changing "The picture is less grim regarding bottom property biases" to "CMIP6 water property biases at the bottom of the North Atlantic are smaller than those at the bottom of the Southern Ocean".
12. L298. Here "convenient" seems like the wrong word. This reviewer can't guess at what to suggest for a replacement.

13. L329-330. Consider changing "are rather accurately represented" to "often agree within observational uncertainties".

14. L469-470. This is a problematic sentence, because NADW does not occupy the global ocean, and in the real world the signatures of upper (subpolar gyre) and lower (GIN sea overflow) NADW are both traceable for substantial distances from their formation regions. How about "NADW formed in the subpolar gyre of the models clearly spreads southward, but the signature of the portions formed in the Nordic seas is less evident."?

Minor comments (typos, debatable word choices, and grammatical errors):

15. L3. Add comma to "transport, and".

16. L7 & L8. Change to "the colder the AABW" & "the saltier the NADW".

17. L10. Change "who" to "which".

18. L25. Change "saltiest" to "saltier".

19. L26. Change "when" to "where"

20. L28. Consider changing "reveal" to "suggest".

21. L29-30. Change "to the bottom of the ocean" to "to the abyssal and deep ocean".

22. L41. Replace the "..." with a ",".

23. L45. Delete "in fact".

24. L77. Delete "note that".

25. L165. Change "who" to "which".

26. L233. Change "convects" to "convect".

27. L373. Change "least" to "less".

28. L402. Change "This sentence is" to "These findings are".

29. L421. Change "blame on" to "attribute to".

---

## Referee Comment (RC2) · Anonymous Referee #2 · 17 Aug 2020

The paper presents a comparison of time mean fields between new, CMIP6 models and observations. The focus is the formation and distribution of deep water formed in the North Atlantic and Southern Ocean. The aim (I assume) is to document these biases as a basis for further study. In general, I did not feel that the paper presented much that was convincing in terms of new scientific interpretation. In particular many of the reported correlations seem small given that the many of the models are not very independent, which I don't think has been accounted for or even acknowledged. Nonetheless, the potential lack of new understanding is not necessarily a strong negative as documenting the models can be a useful exercise in and of itself.

A secondary issue I had with the paper was the overall style, which I found to be far too colloquial. Many examples are provided below, though this is not an exhaustive list.

[Figure]

Scientific issues:

L77-78. Why is a different threshold used to the observations? How can you then fairly compare with the observations? Please explain this.

L224-226. So what can we actually determine or learn from this? Is there a relationship, particularly after controlling for the fact that several of the models are nearly identical (or assimilate observations, which I am surprised is included as it seems fundamentally different to the other models)

L250-251. This link surely only makes sense if the climate sensitivity is driving the DMV. But previously you suggest the logic is the other way around (i.e. larger DMV can sequester more heat and thus reduce climate sensitivity). If the DMV drives the climate sensitivity, why should DMV in the Weddel Sea and SPG themselves be linked? Isn't it more likely that the DMV in these two regions correlate due to some global model bias?

L364, L367. Are these correlations really robust given the real number of degrees of freedom is likely far fewer than the total number of models

Minor or style issues:

L1. "Deep water formation is the driver of the global ocean circulation" - on what timescale?

L5. Large majority - can you be more quantitative?

L5. "Pipe" - I'm not sure that quotation marks are useful here, just leave it as "overflow parameterisation"

L7, L8. Should be a "the" before AABW, NADW

L10. Who - replace with "that"

L11. Also are - are also

L14. Don't start a sentence with "but"

L25. Quits - leaves

L28. These quotation marks and the equals sign are very colloquial.

L31. Is an accurate representation really needed for climate predictions? Over what timescales are you referring? Of what variable? Predictions in CMIP usually refers to initialised decadal simulations, whereas projections usually refer to century timescale uninitialised simulations.

L35. Occurring-too-often - overly frequent

L38. Too - overly

L41. Why are there ellipses here?

Table 1. The horizontal resolution here doesn't take into account any local grid refinement, which could also be noted.

L58. Is this the actual variable name or is it mlotst?

L58. For the models where you have MLD directly, can you show as a supplementary figure that your method and the online one are equivalent.

L72. Is the deboyer montegut MLD data just to 2004 or is it updated?

L77. Detected - diagnosed

L91. "Really" is not required

L104. "Hardly a third" is colloquial. Please be specific.

L109. Starting a sentence with "but"? Multiple other occurrences.

L164. Re-occurring too often - too frequent

L168. "... is but a..." - is just

Interactive
comment

L173. "(Thin?)" - this made me a bit annoyed. If you want to hypothesize at the reason then please spell this out in a sentence rather than like this.

L182. "Thrice" - three times

L182. Erroneous period after "not"

L183. Does the pipe physically suck the water in the model physics? Could the description of this parameterization be described more formally?

L185. What is "the mystery remains" doing in a scientific paper?

L185. "Maybe it is formed..." Remove this and combine with the previous sentence.

L185. Conclusion needn't be pluralized

L205. Regionally varying biases perhaps?

L212. Biases as rmse... What does this sentence mean?

L228. "Tug of war" - this seems to anthropomorphize the models and its not clear what it is trying to convey. Perhaps "balance" would be better?

L233. Convects needn't be pluralized

L243. Here (and elsewhere) the referencing is a bit unclear. You've already cited this paper, and here it seems as if you're citing it as a reference showing a link between convection and climate sensitivity here.

L254. "Besides,..." - please make this sentence better

L257. Please define quantitatively what a "tolerable" bias is?

L265. Why is the word "grim" in a scientific paper?

L267. Don't start sentence with And

L286. Missing "the" before SPG
Interactive
comment

L289. Is such a small correlation actually significant, especially when accounting for the limited degrees of freedom (i.e. many similar models)

L297. "The question remains". What question? Please specify.

L298. "Convenient" seems to be assigning higher purpose to these correlations, which can't be right.

L305-307. Missing "the" before SPG/GIN

L320. Don't start sentence with "which"

L321. The AMOC can't be said to be overestimated given the quoted uncertainty on both the model and observations.

L330. Why is there an exclamation mark?

L335. Remove "obviously"

L341. Missing "of"

L402. "This sentence is still valid" - "this is still the case"

L403. Why is there a floating question mark?

L427. Sentence starting with "which"

---

## Author Comment (AC1) · 15 Sep 2020

The author sincerely thanks Reviewer 1 for their really helpful comments: not only did they point out issues, they also constructively suggested solutions. The role of the reviewers has been duly acknowledged:

L.554-555: "The author thanks the two anonymous reviewers whose comments greatly improved the quality of this manuscript"

The response is organised as follows: first, the reviewer's comment. Then, my answer. Finally, if relevant, new or modified text (with line number for additions)

1. While the 6x6 postage stamp figure format is quite useful, especially for the modelers, it may be a bit much for some readers. Please consider putting all the postage

stamp figures into the appendix and switching the text figures to 3-panel figures (water property reference, mean or median difference of water property from reference, std deviation or interquartile range of water property difference from reference). This would keep 4 figures in the main text: Two 3x3 figures (3 panels each of reference, mean, and standard deviation for potential temperature, practical salinity, and sigma-4) in the S. Ocean and N. Atlantic. Two 1x3 figures of reference, mean differences, and standard deviations of AABW thickness and NADW thickness This change would mean eight postage stamp "difference" figures in the appendix (NADW thickness, AABW thickness, and temper, salinity, and sigma-4 for both the Southern Ocean and North Atlantic. This suggested change would also require some additional text to describe the means and standard deviations of the differences of the models from observations, but it might be worth it. The statistical figures would probably be much more widely used than the postage stamp figures. They would be great in a general presentation on climate model evaluation.

This comment from the reviewer echoes the first comment made by reviewer 2, that the target or even the aim of this paper was somewhat unclear. This paper was written with model users in mind, with the aim to be the go-to reference when they need to justify their choice of models for their study. As I had to do at that time for another study with the CMIP6 models. Consequently, I am reluctant to send the postage stamp figures to the appendix, as at least to me they are the most useful. Nevertheless, I also really liked the reviewer's suggestion of what could be called "summary figures". So I created them, and in fact expanded on the reviewer's idea to make it two 4 x 4 figures with reference, multi model mean, mean difference in property from reference, and standard deviation, for density, temperature, salinity and mixed layer depth, in the Southern Ocean and North Atlantic, complete with the median value quoted on each panel. As the aim of this change is to make the paper useful to as wide a range of readers as possible, I decided to include both the multi model mean and the mean difference in properties, even though they show the same thing. These two figures now are in the text as (new) Figs. 2 and 4, and their description has been added in

the text where relevant. I chose to not add such figures for the AABW and NADW thickness, as the results of these postage stamps figures are already summarised in Tables B1 and B2.

Major comments:

2. L1. The first sentence is arguably inaccurate, as the global ocean circulation is mostly wind driven. Maybe "Deep and bottom water formation are an important part of the global ocean circulation" would be better.

Sentence changed to: "Deep and bottom water formation are crucial components of the global ocean circulation"

3. Throughout. Please use the commonly accepted terminology "bottom water" to distinguish Antarctic Bottom Water from "deep water" (North Atlantic Deep Water). The manuscript may be slightly longer as a result (as sometimes the use of "deep and bottom water" will be required), but will be more in line with the field, and easier to read.

Terminology changed throughout the manuscript to: "deep and bottom water" when referring to both AABW and NADW in the same sentence; "bottom water formation" when referring to AABW only.

4. L19. The opening sentence of the introduction doesn't seem accurate, and it is difficult to parse the meaning. How about "Bottom water formation around Antarctica and deep water formation in the North Atlantic ventilate the global abyssal and deep ocean."?

Sentence modified as suggested.

5. L23. This sentence does not seem quite right, as AABW influence spreads quite far into the N. Atlantic, albeit in a form highly diluted by mixing with NADW above. How about "In a substantial portion of the Atlantic, Antarctic Bottom Water spreading north is overlain by North Atlantic Deep Water spreading south."

Sentence modified as suggested.

6. L63-65. This sentence conflates two different issues, and perhaps should be broken in two at the "and".

Sentence modified to clarify how the two issues are related: "Furthermore, as some models are not fully independent as they share similar codes (Table 1), using different ensemble sizes would have accentuated the bias towards one model family"

7. L140-142. This portion is overly concise. How about something along the lines of: "Presently, Antarctic Bottom water is primarily formed in several locations (including the Weddell Sea, the Ross Sea, and the Adelie Lands) as water is cooled, made saltier,and denser on the continental shelves, then cascades down the continental slopes,entraining deep waters on its way to the sea floor."

The sentence suggested by the reviewer has been added.

8. L160-161. The recent Polynya in the Weddell Sea that is mentioned later in the paper should also be mentioned here.

Sadly, the recent Maud Rise polynya was not associated with any deep convection (e.g. Campbell et al., 2019), so it cannot be mentioned here.

9. Tables 2 and 3. Consider that instead of years out of 30, a percentage would be more quickly and easily interpreted by most readers.

I agree with the reviewer that it would be quicker and easier, but it would also be misleading. The CMIP6 runs contain more than 100 years; presenting these values as a percentage would imply for many readers that I worked with 100 years. Hence I prefer keeping the values in the table as "years out of the 30 studied".

10. L205. How is it possible that "No model has regional biases"? Perhaps just delete the part of this sentence on L206, and start the sentence with "CMIP6 models tend to be either biased light or biased dense".

[Figure]

**OSD**

Here I tried to mean that unlike some models in CMIP5, there is no difference in the sign of the bias between e.g. the Weddell and Ross seas. I cannot find a better reformulation than what the reviewer suggests, so the sentence was modified as suggested.

11. L265. Consider changing "The picture is less grim regarding bottom property biases" to "CMIP6 water property biases at the bottom of the North Atlantic are smaller than those at the bottom of the Southern Ocean".

Sentence modified as suggested.

12. L298. Here "convenient" seems like the wrong word. This reviewer can't guess at what to suggest for a replacement.

The word has been removed: "a strong interhemispheric correlation"

13. L329-330. Consider changing "are rather accurately represented" to "often agree within observational uncertainties".

The agreement with observational uncertainties is mentioned several times in the previous sentences. The last sentence "in summary, [they] are rather accurately represented" stays.

14. L469-470. This is a problematic sentence, because NADW does not occupy the global ocean, and in the real world the signatures of upper (subpolar gyre) and lower (GIN sea overflow) NADW are both traceable for substantial distances from their formation regions. How about "NADW formed in the subpolar gyre of the models clearly spreads southward, but the signature of the portions formed in the Nordic seas is less evident."?

Sentence modified as suggested

Minor comments (typos, debatable word choices, and grammatical errors): All addressed

[Figure]

**Figure 2.** Multi model summary of the biases for the Southern Ocean. Columns: observational reference and multi model mean property (same colorbar for both), mean bias (model minus reference) and standard deviation of the difference; rows: bottom density, bottom potential temperature, bottom salinity and mixed layer depth. White numbers are the median value over the deep Southern Ocean, defined as per Fig. 1. Grey lines indicate the 1000 m and 2000 m isobaths.

**Fig. 1.** New figure suggested by the reviewer: Southern Ocean

[Figure]

**Figure 4.** Multi model summary of the biases for the North Atlantic. Columns: observational reference and multi model mean property (same colorbar for both), mean bias (model minus reference) and standard deviation of the difference; rows: bottom density, bottom potential temperature, bottom salinity and mixed layer depth. White numbers are the median value over the deep GIN (top) and SPG (bottom), defined as per Fig. 3. Grey lines indicate the 1000 m and 2000 m isobaths.

**Fig. 2.** New figure suggested by the reviewer: North Atlantic

[Figure]

---

## Author Comment (AC2) · 15 Sep 2020

The author thanks Reviewer 2 for their scientific comments. The role of the reviewers has been duly acknowledged: L.554-555: "The author thanks the two anonymous reviewers whose comments greatly improved the quality of this manuscript"

The response is organised as follows: first, the comment from the reviewer; then, my answer; finally, when relevant, new or modified text.

The paper presents a comparison of time mean fields between new, CMIP6 models and observations. The focus is the formation and distribution of deep water formed in the North Atlantic and Southern Ocean. The aim (I assume) is to document these biases as a basis for further study. In general, I did not feel that the paper presented much that

was convincing in terms of new scientific interpretation. [. . .] Nonetheless, the potential lack of new understanding is not necessarily a strong negative as documenting the models can be a useful exercise in and of itself.

Both reviewers raised this important point: the target audience and aim of the paper were not as clearly defined as I thought. This paper is indeed designed as a reference for model users to justify their choice of models for further studies. Although attempts are made at explaining these biases, the emphasis is on quantifying these biases. This sentence was added to the introduction to clarify this objective, lines 51-52: "The primary objective of this paper is to quantify and discuss biases of each model, so that model users can make informed model selections."

In particular many of the reported correlations seem small given that the many of the models are not very independent, which I don't think has been accounted for or even acknowledged.

The lack of independence of the models is acknowledged as early as line 67 (slightly modified from the previous version in response to a comment by reviewer 1): "Furthermore, as some models are not fully independent as they share similar codes (Table 1). . ." In response to the reviewer's comment, this methodological clarification was also added line 68: "To account for this lack of independence, the correlations quoted throughout the text have been verified with different model numbers"

Scientific issues:

L77-78. Why is a different threshold used to the observations? How can you then fairly compare with the observations? Please explain this.

I chose neither threshold. The threshold of the models is the "official" threshold of the CMIP6 procedure; that is, models that wanted to participate in CMIP6 had to use that threshold. Likewise, the threshold used in observations is the one that was chosen by the creators of this observational product. The literature on the impact of one threshold

rather than another is plentiful (I even wrote a PhD dissertation chapter on this), and the conclusion is that for detecting spurious modelled deep convection, this difference is not critical. In fact, choosing a larger threshold for the models than for the observations means that we would underestimate deep convection in models. As the objective of this publication is primarily to compare models with each other, the most important is that all models use the same threshold. The following was added to summarise this discussion, lines 80-88: "As is requested for CMIP6, the MLD is then detected as the depth where $\sigma\theta$ differs from that at 10 m depth by more than 0.125 kg m$-3$. [new text associated with the new supplementary figure] Furthermore, a different threshold of 0.03 kgm$-3$ is used in the observational reference (de Boyer Montégut et al., 2004), which could lead to an underestimation of mixed layer depths in the models (as we show in section 3, it does not)."

L224-226. So what can we actually determine or learn from this? Is there a relationship, particularly after controlling for the fact that several of the models are nearly identical (or assimilate observations, which I am surprised is included as it seems fundamentally different to the other models)

Sentence expanded to make the point clearer: "The models that convect the least or not at all tend to be the most accurate. For the CESM2 family, accurate bottom properties and lack of deep convection may both be the result of their overflow parameterisation (Briegleb et al., 2010; Snow et al., 2015). For another model, NorCPM1,the accuracy in all properties may come from its observation assimilation rather than accurate model physics (Counillon et al., 2016)."

L250-251. This link surely only makes sense if the climate sensitivity is driving the DMV. But previously you suggest the logic is the other way around (i.e. larger DMV can sequester more heat and thus reduce climate sensitivity). If the DMV drives the climate sensitivity, why should DMV in the Weddell Sea and SPG themselves be linked? Isn't it more likely that the DMV in these two regions correlate due to some global model bias?

Does the DMV impact the climate sensitivity, or does the sensitivity impact the DMV? Less mixing means less heat absorbed by the ocean, so more in the atmosphere and a larger sensitivity. The opposite is true: if the sensitivity is somehow controlled by another "global model bias", a high sensitivity will lead to more ocean surface warming and stratification, and hence less mixing. As is obvious from just these two sentences, what we really have is a feedback loop, and investigating which comes first, or what that other global bias can be, are beyond the scope of this paper. I added a sentence to reflect the point raised by the reviewer lines 277-279: "As already mentioned, no causation can be inferred: deciphering whether global biases in DMV are responsible for the models' sensitivities, or in contrast sensitivities are set by other processes and impact the DMV, is beyond the scope of this analysis"

L364, L367. Are these correlations really robust given the real number of degrees of freedom is likely far fewer than the total number of models

All the correlations have been verified using different model numbers, in particular using only one member per family, and the results remained. This precision has been added to the Methods section, lines 68-69.

Minor issues:

To summarise, all the issues highlighted by the reviewer that impeded the understanding have been corrected. Only the ones for which a response longer than "corrected" was necessary are presented here. I leave it to the copy editor to decide whether grammar rules that we learnt at school can be bent in order to make the manuscript more dynamic and pleasant to read.

L1. "Deep water formation is the driver of the global ocean circulation" - on what timescale?

This sentence has already been modified in response to a comment by Reviewer 1.

L5. Large majority - can you be more quantitative?

[Figure]

Sentence modified to "28 models in the Southern Ocean and all 35 models in the North Atlantic"

L31. Is an accurate representation really needed for climate predictions? Over what time scales are you referring? Of what variable? Predictions in CMIP usually refers to initialised decadal simulations, whereas projections usually refer to century time scale uninitialised simulations.

I am not sure which point the reviewer is trying to make here. I changed "prediction" to "projection", as I meant long term, IPCC-report type results.

Table 1. The horizontal resolution here doesn't take into account any local grid refinement, which could also be noted

They could indeed, but too many cases would need to be considered to fit in the table. Instead, table caption has been modified to "nominal" grid resolution.

L58. Is this the actual variable name or is it mlotst?

The actual variable name is mlotst. There was a typo.

L58. For the models where you have MLD directly, can you show as a supplementary figure that your method and the online one are equivalent.

This is a very good idea. A new supplementary Figure (A1) has been added to show where and by how much they differ, along with a discussion in the Methods section.

L72. Is the deboyer montegut MLD data just to 2004 or is it updated?

It is updated, as is indicated on the data download website.

L104. "Hardly a third" is colloquial. Please be specific.

The number of models changed since the initial submission. Sentence now reads: "it is provided by only 18 of the models (from 10 families)"

L173. "(Thin?)" - this made me a bit annoyed. If you want to hypothesize at the reason

then please spell this out in a sentence rather than like this

I am confused by the reviewer's reaction as the hypothesis is spelt out in that sentence. I removed the "(thin?)" that brought nothing to the sentence.

L183. Does the pipe physically suck the water in the model physics? Could the description of this parameterization be described more formally?

The following sentence was added lines 196-198: "If the water on the shelf exceeds a critical density, a pipe artificially transports this dense water from the shelves to the deep basin. Without having to cascade, the dense shelf water keeps its properties."

L243. Here (and elsewhere) the referencing is a bit unclear. You've already cited this paper, and here it seems as if you're citing it as a reference showing a link between convection and climate sensitivity here.

I assume the reviewer meant the reference to Zelinka et al. (2020) of (previous version's) line 248. Sentence modified to clarify that I refer to this paper as the source of the sensitivity values I used: "There is a relationship with the climate sensitivities of Zelinka et al. (2020) though"

L257. Please define quantitatively what a "tolerable" bias is?

I do not understand what the reviewer means as the quantitative value is given in the same sentence, two words later.

L289. Is such a small correlation actually significant, especially when accounting for the limited degrees of freedom (i.e. many similar models)

This point has been addressed twice already in this response.

L297. "The question remains". What question? Please specify.

Sentence now reads: "the cause of NorCPM's and other models' bottom density bias in the GIN seas remains unknown"

L321. The AMOC can't be said to be overestimated given the quoted uncertainty on both the model and observations.

Agreed. Sentence removed.

[Figure]

[Figure]

**Figure A1.** Maximum monthly mixed layer depth in the North Atlantic over 1985-2014 for the model CanESM5: left, using the model output 'mlotst'; right, when computed from the monthly temperature and salinity. Over the entire 30-year period, the root mean square error in the Nordic Seas is 305 m; in the subpolar gyre, 21 m.

**Fig. 1.** New supplementary figure suggested by the reviewer: comparison of the MLDs

---

## Author Response (AR2)

The author is grateful to the reviewer for their new comments.

As previously, the reviewer's comments are in **bold font**, followed by the author reply in plain font and, when relevant, corresponding changes to the text in *italics*.

**Substantive suggestions:**

**1. L23-24. The phrase "2000 m thick" is a bit too precise here. Perhaps "several hundred to few thousand meter thick" might be more accurate?**

Sentence changed as suggested:
*"the Antarctic Bottom Water (AABW). AABW does not stay around Antarctica, but rather travels north on the sea floor as a several hundred to few thousand meter thick layer, filling all three basins"*

**2. L83. Here is it not "the non-linearity of the equation of state" but probably much more owing to the positive skewness of mixed layer depth distributions. If one estimates mixed layer depths from instantaneous, or even daily profiles of water properties, in many locations one will obtain many shallow values and a few deep values. If one uses monthly average profiles they are likely to result in a deeper mixed layer than the average of the daily or hourly values.**

I think both the reviewer and I mean the same thing, i.e. that it is an artefact from the monthly averaging of temperature and salinity. I still think that the difference would be less pronounced if the EOS were linear. The text has been changed to clarify this:
*"Because of averaging effects, potentially combined with the non-linearity of the equation of state, this MLD computed from monthly temperature and salinity differs slightly from mlotst"*

**3. L105 and following. The GIN sea component of the NADW is confusing. It seems like in this paper the GIN seas component is estimated within the GIN seas, whereas in Johnson (2008) it is estimated from near-bottom data in the Northern N. Atlantic, after most of the entrainment during the overflow process has taken place. Therefore the GIN seas component of NADW in Johnson (2008) will be much more "Atlantic" in character than in this paper. Does this cause issues? At the very least the difference should be pointed out in the revised manuscript.**

This is a very valid comment. The difference in method between the real ocean and the models is motivated by past studies (notably by the author) that showed biases in the representation of properties and formation of both North Atlantic water masses, along with not shown tests on these CMIP6 models. As suggested by the reviewer, we now point out the difference in the methods and justify it as explained here:
*"Note that the definition of NADW$_{GIN}$ is different from that of Johnson (2008) because of extra tests not shown here, and known past model biases in the vertical structure of the North Atlantic (e.g. Menary and Wood, 2018) and in the North Atlantic – Nordic Seas exchanges (e.g. Heuzé and Årthun, 2019)."*

**4. Figures 2 and 4. If the colorbar were flipped for the standard deviation plots so that lighter shades correspond to smaller magnitudes, and darker shades to larger magnitudes, this reviewer**

**would have an easier time assessing the patterns as the regions with large standard deviations would stand out.**

Black means 0 both in RGB and in albedo, so personally, I expect black to mean zero when I see it on any colour map. On this occasion, black is also the colour of the land on the other maps (as in Johnson 2008), and the land has a standard deviation of 0. If either the editor or the copy-editor insists, the colour map can be changed, but it is not changed here in this version.

**5. L318. Change "fresh" to "light"? Density is not fresh, it is either light or dense.**

It was a typo, thank you for noticing it. It now reads "light".

**Minor suggestions:**

**6. L153. The quotation marks are unnecessary here.**

Agreed, removed.

**7. L181. Change "who" to "which".**

Changed.

**8. L368. Change "between" (which is used when comparing two things) to "among" (which is used when comparing more than two things).**

Changed.

[revised manuscript text omitted]

---

## Author Response (AR3)

I would like to thank the editor for his comments and support. On this last version, the editor made one typographic comment:

"line 298: magnitude than those -> magnitude as those"

I confirm that I corrected the text.

I do not attach to this document the track-change version of the manuscript.